# Transcriptomic complexity of the human malaria parasite *Plasmodium falciparum* revealed by long-read sequencing

**Philip J. Shaw**[1]◉*, **Pavita Kaewprommal**[2], **Chayaphat Wongsombat**[1], **Chumpol Ngampiw**[2], **Tana Taechalertpaisarn**[3], **Sumalee Kamchonwongpaisan**[1], **Sissades Tongsima**[2], **Jittima Piriyapongsa**[2]◉*

**1** National Center for Genetic Engineering and Biotechnology (BIOTEC), National Science and Technology Development Agency (NSTDA), Pathum Thani, Thailand, **2** National Biobank of Thailand (NBT), National Science and Technology Development Agency (NSTDA), Pathum Thani, Thailand, **3** Department of Microbiology, Faculty of Science, Mahidol University, Bangkok, Thailand

◉ These authors contributed equally to this work.
* philip@biotec.or.th (PJS); jittima.pir@nstda.or.th (JP)

**Data Availability Statement:** All raw data associated with the manuscript are now publicly available from NCBI GEO via accession number

## Abstract

The *Plasmodium falciparum* human malaria parasite genome is incompletely annotated and does not accurately represent the transcriptomic diversity of this species. To address this need, we performed long-read transcriptomic sequencing. 5′ capped mRNA was enriched from samples of total and nuclear-fractionated RNA from intra-erythrocytic stages and converted to cDNA library. The cDNA libraries were sequenced on PacBio and Nanopore long-read platforms. 12,495 novel isoforms were annotated from the data. Alternative 5′ and 3′ ends represent the majority of isoform events among the novel isoforms, with retained introns being the next most common event. The majority of alternative 5′ ends correspond to genomic regions with features similar to those of the reference transcript 5′ ends. However, a minority of alternative 5′ ends showed markedly different features, including locations within protein-coding regions. Alternative 3′ ends showed similar features to the reference transcript 3′ ends, notably adenine-rich termination signals. Distinguishing features of retained introns could not be observed, except for a tendency towards shorter length and greater GC content compared with spliced introns. Expression of antisense and retained intron isoforms was detected at different intra-erythrocytic stages, suggesting developmental regulation of these isoform events. To gain insights into the possible functions of the novel isoforms, their protein-coding potential was assessed. Variants of *P. falciparum* proteins and novel proteins encoded by alternative open reading frames suggest that *P. falciparum* has a greater proteomic repertoire than the current annotation. We provide a catalog of annotated transcripts and encoded alternative proteins to support further studies on gene and protein regulation of this pathogen.

GSE186109. Please check the following link to data: https://www.ncbi.nlm.nih.gov/geo/query/acc.cgi?acc=GSE186109.

**Funding:** This work was supported by the Platform Technology Management section, National Center for Genetic Engineering and Biotechnology (BIOTEC, Thailand) grant P1551103 (P.J.S & J. P), and the National Science and Technology Development Agency, (Thailand) grant P1850116 (Research Chair Grant) and P1450883 (S. K.). The funders had no role in study design, data collection and analysis, decision to publish, or preparation of the manuscript.

**Competing interests:** The authors have declared that no competing interests exist.

## Introduction

Malaria remains a common parasitic disease in humans, with over 200 million cases in 2020 [1]. The majority of malaria cases are attributed to *Plasmodium falciparum* infection. *P. falciparum* belongs to the protist phylum Apicomplexa, which includes many other disease-causing parasites. Motile *P. falciparum* sporozoites infect humans via the bites of *Anopheles* spp. mosquitoes. The parasites initially invade human liver cells, where they multiply and then egress into the bloodstream as merozoites capable of invading mature erythrocytes. Parasites reproduce asexually inside erythrocytes during the approximately 48 h intraerythrocytic development cycle (IDC), passing through the morphologically distinct ring, trophozoite, and schizont developmental stages. IDC stages are amenable to continuous in vitro culture [2], which has facilitated molecular studies of the parasite.

A major landmark in the understanding of *P. falciparum* biology was the sequencing of the genome [3]. The *P. falciparum* genome is gene-dense in common with other apicomplexan parasite species [4]. Approximately half of *P. falciparum* annotated genes have introns. Most annotated splicing junctions in *P. falciparum* genes have been validated by short-read RNA-Seq data. Isoform events, including retained introns, exon-skipping, and alternative 5′ and 3′ splicing junctions, have been described [5–8]; however, the true level of isoform events in *P. falciparum* is thought to be substantially higher [9].

The untranslated regions (UTRs) flanking open reading frames (ORFs) of *P. falciparum* protein-coding genes are extremely AT-rich, low-complexity sequences, which have hampered comprehensive annotation of transcripts. De novo assembly of transcripts from RNA-Seq data [10–12] revealed long UTRs consistent with earlier estimates from northern blotting [13]; however, gaps in RNA-Seq coverage led to the erroneous assignment of many UTRs as novel long non-coding RNA (lncRNA) [14]. To overcome the problem of discontinuity in RNA-Seq coverage, RNA-Seq data obtained using an amplification-free approach helped to map the furthest extents of UTRs with much greater accuracy [15]. The current *P. falciparum* 3D7 nuclear genome annotation (PlasmoDB [16] release 58) includes UTR features for the majority of the 5,318 protein-coding genes (not including pseudogenes) based on these data. 67(1%) of the genes have annotated alternative transcripts; however, RNA-Seq studies using specialized protocols to enrich fragments of 5′ capped transcript ends have indicated that isoforms with alternative 5′ ends showing differential abundance across the IDC are expressed from many more genes [17,18].

Long-read, single-molecule transcriptomic sequencing using Pacific Bioscience (PacBio), and Nanopore platforms can provide full-length isoform sequences. Data from the PacBio platform are more accurate than Nanopore owing to the lower error rate; however, Nanopore experiments can yield substantially more data. The full-length isoform sequences from long-reads reveal a greater diversity of isoform events and transcript structures than short-read RNA-Seq [19,20]. Long-read RNA-Seq data have been reported for *P. falciparum* using the PacBio [15,21] and Nanopore platforms [22]. Despite the availability of these data for *P. falciparum*, no attempt has been made to comprehensively annotate isoforms and associated isoform events, particularly alternative ends.

In this study, we generated long-read RNA-Seq data for *P. falciparum* IDC stages. Samples of total and nuclear-enriched parasite RNA were obtained and subjected to enrichment for 5′ capped mRNA. The mRNA samples were converted to cDNA and sequenced using the PacBio and Nanopore platforms. Candidate non-redundant isoforms were identified in each dataset, together with long-read RNA-Seq data from previous studies. A comprehensive isoform annotation was created from the available data and isoform events were characterized in detail.

## Results

### Long-read cDNA sequencing and construction of the transcriptomic catalog

Previous long-read RNA-Seq studies of the *P. falciparum* transcriptome were performed with total RNA [21] or polyA-enriched mRNA [15,22]. As no 5′ end enrichment was performed in these studies, sequences originating from degraded mRNA lacking a 5′ cap could obscure the identification of isoforms with alternative 5′ capped ends. To increase the representation of isoforms with alternative 5′ ends, we enriched 5′ capped mRNA from samples of *P. falciparum* IDC stages using a 5′ cap-binding protein as described in previous studies [12,23]. For this study, we used recombinant human eIF4E fused to a fragment of human eIF4G [24]. The fusion protein demonstrated 5′ cap-binding activity and was purified for enriching 5′ capped mRNA (S1 Fig). In addition to 5′ capped mRNA enrichment, we sought to increase the representation of nuclear isoforms by subcellular fractionation. Nuclear localization of *P. falciparum* mRNAs has been reported, including antisense lncRNA regulators of *var* genes [25]. The purity of isolated nuclei was assessed by confocal microscopy, which revealed intact nuclei separated from other cellular structures (S2 Fig).

5′ cap-enriched mRNA samples were used to construct cDNA libraries for sequencing on Nanopore and PacBio platforms. The Nanopore platform has a higher base-calling error rate than that of the PacBio platform, which could lead to incorrect annotation, in particular, if inappropriate analysis methods are used. RNA sequin synthetic RNAs [26] were spiked into the *P. falciparum* RNA samples for the construction of cDNA libraries for Nanopore sequencing, and we used the RNA sequin data to identify the most suitable data analysis method. The first step of transcriptome annotation involves clustering long reads to generate candidate non-redundant isoform sequences. Technical noise, including truncated fragments and sequencing errors, can lead to spurious candidate isoforms from clustering [27]. The noise can be identified if ground-truth transcriptomic annotation is available. From the comparison of candidate isoforms with annotation, all tools assigned more isoforms to classes other than the ground truth class, with the majority of incorrectly assigned isoforms belonging to truncated fragment classes (Fig 1A). Although the StringTie2 [28] and FLAIR [29] tools were more sensitive than TAMA collapse with a greater number of ground truth transcripts reported, String-Tie2 and FLAIR falsely assigned isoforms from misaligned reads (identified from isoforms mapping to non-exonic regions including antisense). We selected the TAMA collapse tool [27] with low stringency settings for assigning *P. falciparum* isoforms because it is less likely to assign false isoforms from misaligned reads with sequence errors than other tools, while it is able to detect most of the expressed isoforms.

Candidate non-redundant *P. falciparum* isoforms were assigned from available data. Summary statistics of read processing, alignment, and candidate non-redundant isoforms for each dataset together with details of sampled developmental stage, RNA fraction, and sequencing platform are provided in S1 Table. The new data generated in this study comprise a greater number of mapped reads and candidate isoforms than those from previous long-read transcriptomic studies combined. Candidate isoforms from all datasets were merged (S3 Fig). Novel isoforms (isoforms not currently annotated as transcripts in PlasmoDB [16]) were filtered to remove those with non-canonical splicing junctions, lacking supporting evidence of 5′ capped end and identified from only a single sample (S4 Fig). The final transcriptomic catalog included 5,568 reference transcripts and 12,495 novel isoforms (S1 Dataset and S2 Table). 12,440/12,495 (99.6%) novel isoforms are supported by PacBio data (S2B Table) indicating that the exon junctions for most isoforms are defined using data from the more accurate long-read sequencing platform. 9,004 of the 12,495 novel isoforms are supported by data from other

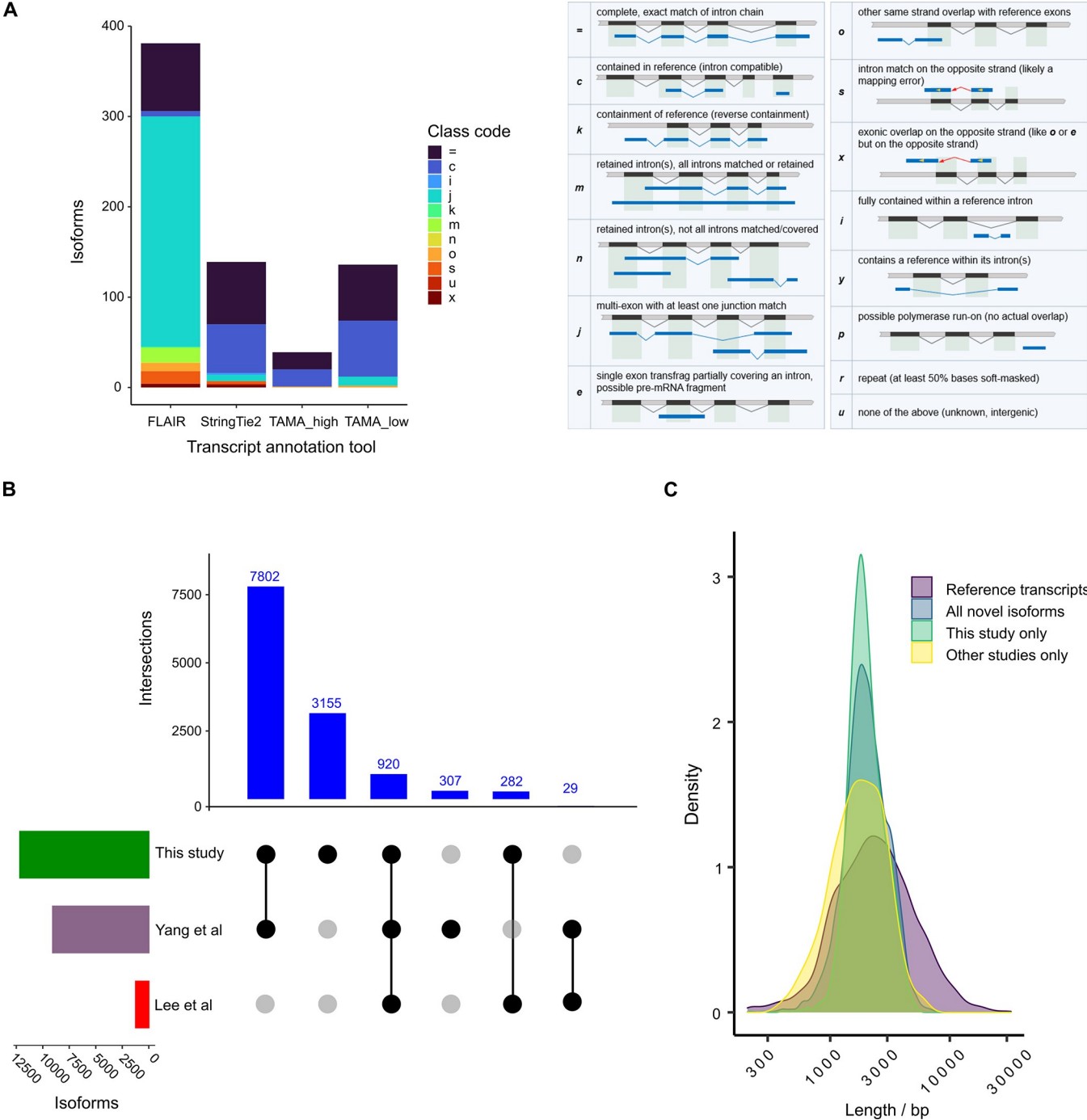

**Fig 1. Isoform annotation and overview of the *Plasmodium falciparum* 3D7 novel isoform catalog. (A)** Comparison of tools for transcript annotation. Candidate non-redundant transcript isoforms representing spike-in synthetic RNAs were identified from the same Nanopore cDNA dataset using three different tools: FLAIR [29], StringTie2 [28], and TAMA collapse [27]. The consistent and inconsistent isoforms outputted by FLAIR were combined. The TAMA collapse tool was run under "low" and "high" stringency settings [27], indicated as TAMA_low and TAMA_high, respectively. Isoforms assigned by each tool were compared with the chrIS annotation [26] using the GffCompare tool [30]. The isoforms matching the annotation (ground truth) are present in the "=" class; other classes not matching the annotation are indicated by the class codes c, k, m, n, j, e, o, s, x, i, and u in decreasing order of priority, i.e., divergence from the ground truth. Class code descriptions are shown on the right (reproduced with permission from [30]). Note that classes c and j include truncated isoforms, s and x include isoforms on the opposite (antisense) strand of the annotated transcript, and class u isoforms map outside of exonic regions. **(B)** Study datasets supporting the catalog of 12,495 *Plasmodium falciparum* 3D7 novel isoforms (not currently annotated as transcripts in PlasmoDB [16]). Intersections are shown by UpSetR plot [31], which is a matrix-based representation of set sizes and their intersections. Matrix rows represent the sets (independent studies) and columns represent intersections of novel isoforms supported by studies. The numbers of isoforms for each study are shown by the

horizontal bar plots on the left. All combinations of set intersections are shown by the matrix cell on the right, in which sets that are part of a given intersection are represented by black-filled circles. Sets that are not part of the intersection are shown as gray circles. The sets considered in each intersection (black circles) are connected by vertical black lines to emphasize the column-based relationships. Vertical bars above the matrix columns represent the number of novel isoforms in each intersection. **(C)** Distributions of transcript lengths as shown by kernel density plot. The distributions of all 12,495 novel isoforms (mean length = 2,163 bp), 3,355 novel isoforms supported by datasets in this study only (mean length = 2,103 bp), 336 novel isoforms supported only by data from other studies [21,22], mean length = 1,942 bp) and 5,568 reference transcripts (mean length = 3,138 bp) are plotted on the same axes. The vertical axis is the probability density function of the variable (isoform length).

studies in combination with data from this study. However, 3,155 novel isoforms are supported by data in this study only, and 336 novel isoforms are supported by data from other studies only (Fig 1B). For further assessment, the length distributions of novel isoforms were compared with reference transcripts (Fig 1C). The mean lengths of all novel isoforms (2.2 kb), novel isoforms unique to this study (2.1 kb), and novel isoforms only from other studies (1.9 kb) are markedly less than that of reference transcripts (3.1 kb). The difference in mean length is expected because the majority of reference transcripts define the longest transcript for each gene.

## Isoform events among novel isoforms

Isoform events were annotated among the catalog of transcripts (S3 Table). Among reference transcripts, the most common events are alternative transcription start site (ATSS) and alternative transcription termination site (ATTS), which represent 82/216 (38%) and 80/216 (37%) of the total, respectively (Fig 2A). Among the events identified in the novel isoforms, ATSS and ATTS are the most common, comprising 10,716/25,056 (43%) and 9,963/25,056 (40%) of the total, respectively (Fig 2B). In contrast to reference transcripts, retained intron (RI) is the next most common event among the novel isoforms (3,599/25,056; 14%). The other types of isoform events are markedly less frequent for both reference and novel transcripts (Fig 2).

The most common isoform event among novel isoforms, ATSS, was investigated in detail. It should be noted that the 5′ ends of 1,535 novel isoforms are within 50 bp of a reference TSS, such that we considered these positions as matching reference TSS. Reference TSS positions correspond to genomic regions of locally elevated H2A.Z epigenetic marks (data from [32]), in which the majority of sites have log-normalized occupancy greater than zero; in contrast, a bimodal distribution is apparent among the 10,323 5′ end positions represented in the novel isoforms (Fig 3A). Two clusters of alternative 5′ end positions were assigned by unsupervised clustering using log-normalized H2A.Z occupancy data. The genomic sequence in the vicinity of reference TSS is characterized by an elevated frequency of thymine upstream and adenine downstream (Fig 3B). Cluster C1 containing the majority of alternative 5′ end positions (7,937) displays a similar genomic sequence pattern (Fig 3C). In contrast, cluster C2 (2,283 5′ end positions) is characterized by an elevated frequency of adenine up and downstream (Fig 3D). The C1 and C2 clusters of alternative 5′ end positions also differ in genomic locations with respect to protein-coding sequence (CDS) regions, in which C1 positions are located mostly outside, whereas C2 positions are mostly within CDS (Fig 3E). As most C2 5′ ends are present within CDS, we wondered if C2 could represent 5′ ends generated post-transcriptionally, perhaps by co-translational cleavage and re-capping as described in other eukaryotes [33]. We examined the frequency of all 64 codons in the vicinity of the C2 5′ ends present within CDS (S5 Fig). Comparison of the distributions by Kolmogorov-Smirnov test showed that 11 codons are significantly different among C2 5′ ends compared with randomly selected nucleotides from the same transcripts (S6 Fig). The codons more frequently associated with C2 5′ ends than random comprise lysine (AAA and AAG), glutamate (GAA and GAG), the arginine codon AGA, and the aspartate codon GAC. The codons less frequently associated with C2 5′

**A**

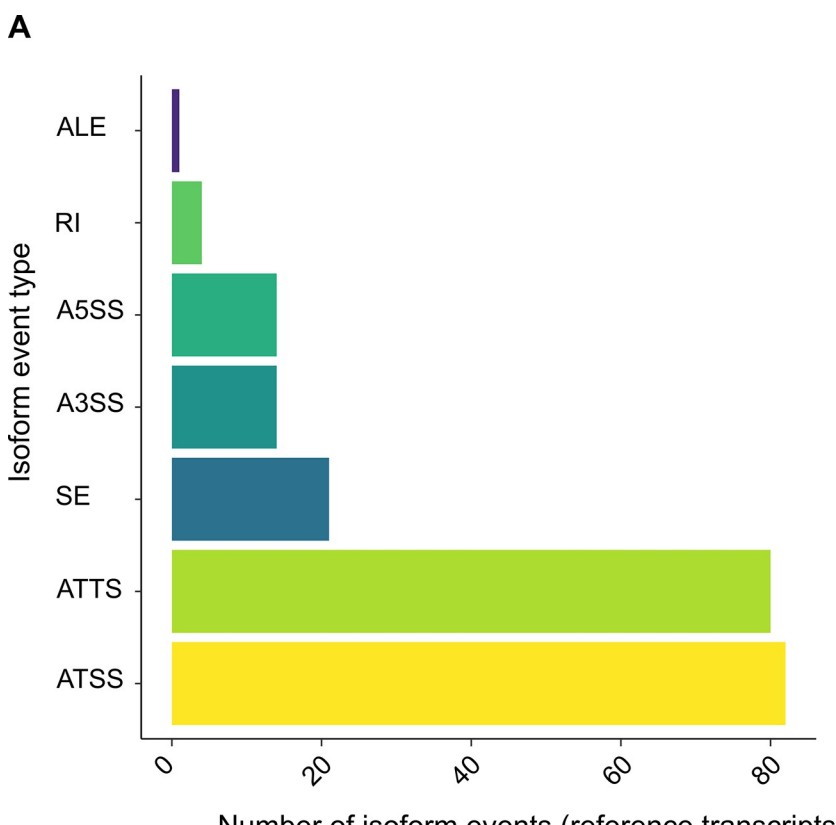

**B**

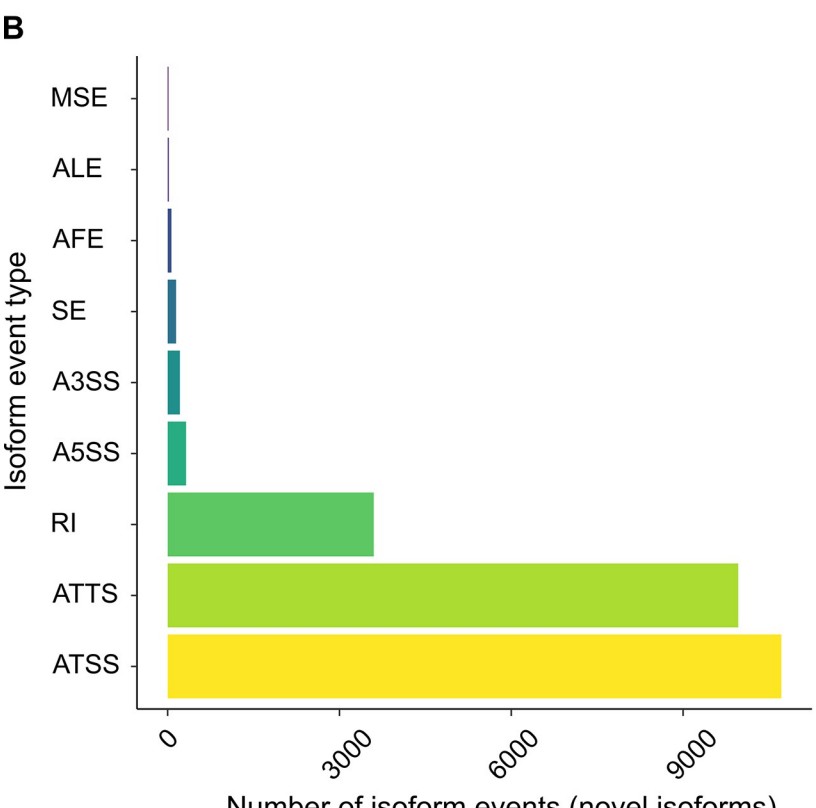

**Fig 2. Isoform events in reference transcripts and novel isoforms.** Isoform events in *Plasmodium falciparum* 3D7 for **(A)** reference transcripts (5,568) and **(B)** novel isoforms (12,495). The different types of isoform events reported include alternative transcription start site (ATSS), alternative transcription termination site (ATTS), skipped exon (SE), alternative 3′ splice site (A3SS), alternative 5′ splice site (A5SS), retained intron (RI), alternative first exon (AFE), alternative last exon (ALE), and multiple skipped exons (MSE).

ends than random comprise TAT (tyrosine), TTT (phenylalanine), ATA and ATT (isoleucine), and TAA (stop).

The next most common isoform event among novel isoforms, ATTS, was investigated in detail. *P. falciparum* TTS positions have been previously mapped from short-read [8,15], and long-read RNA-Seq data [21]. However, to our knowledge, no sequence analysis of TTS, including investigation of polyadenylation signals, has been described for this organism. A high local frequency of adenine is apparent at reference TTS, which extends a short distance and is flanked by a longer distal region with an elevated frequency of thymine compared with random positions (Fig 4A). Inspection of the most-over-represented 6-mer at each position upstream of reference TTS revealed 37 different 6-mers (Fig 4B; full results and statistics in S4 Table), meaning that it is not possible to define a single major polyadenylation signal motif for this species. The 10,753 ATTS represented among the novel isoforms show an even more pronounced adenine-rich signal (Fig 4C). One important caveat for the analysis of ATTS is that these events lack corroborative evidence from feature-enriched short-read RNA-Seq, unlike ATSS. Spurious ATTS events can arise from 3′ truncated sequences generated by template switching and internal priming during cDNA synthesis [34]. In mammalian transcriptomic studies, these artifacts can be eliminated by removing isoforms with 3′ ends mapping to locally adenine-rich genomic regions and considering ATTS as genuine only if canonical polyadenylation signals are present upstream [34,35]. However, the presence of adenine-rich motifs in the vicinity of bona fide reference TTS and the lack of a definitive canonical upstream polyadenylation signal disqualifies this filtering approach for *P. falciparum*. We investigated ATTS with supporting evidence from direct RNA sequencing, which can be used for validation [34]. ATTS supported by direct RNA sequencing (943 positions) show a similar base frequency pattern to all ATTS (Fig 4C).

The next most common type of isoform event, RI, was investigated in detail. Comparison of the percent GC and length distributions by Kolmogorov-Smirnov test showed significant differences between RI and spliced introns, in which RI are skewed towards higher percent GC (Fig 5A) and shorter length (Fig 5B) compared with spliced introns; however, it should be noted that the mean differences are small. Base composition plots of 5′ and 3′ splicing junctions did not reveal marked differences between RI and spliced introns, which show essentially the same pattern of adenine-rich sequence flanking either side of the 5′ junction (Fig 5C) and thymine-rich sequence upstream of the 3′ junction (Fig 5D).

## Expression patterns of retained introns and antisense isoforms

We obtained total and nuclear RNA samples of parasites at different stages of development for constructing PacBio libraries. From these PacBio library datasets, we obtained information on stage and nuclear representation for 11,325 of the novel isoforms in our catalog. We investigated whether different structural categories of the novel isoforms were over-represented exclusively in the nucleus or at different developmental stages by Fisher's exact test (Fig 6). The structural category "novel_in_catalog", and the subcategories "intron_retention" and "mono-exon_by_intron_retention" showed significantly greater odds ratios among isoforms exclusively represented by nuclear datasets. The same structural category and sub-categories showed significantly greater odds ratios for the trophozoite and schizont stage datasets. The

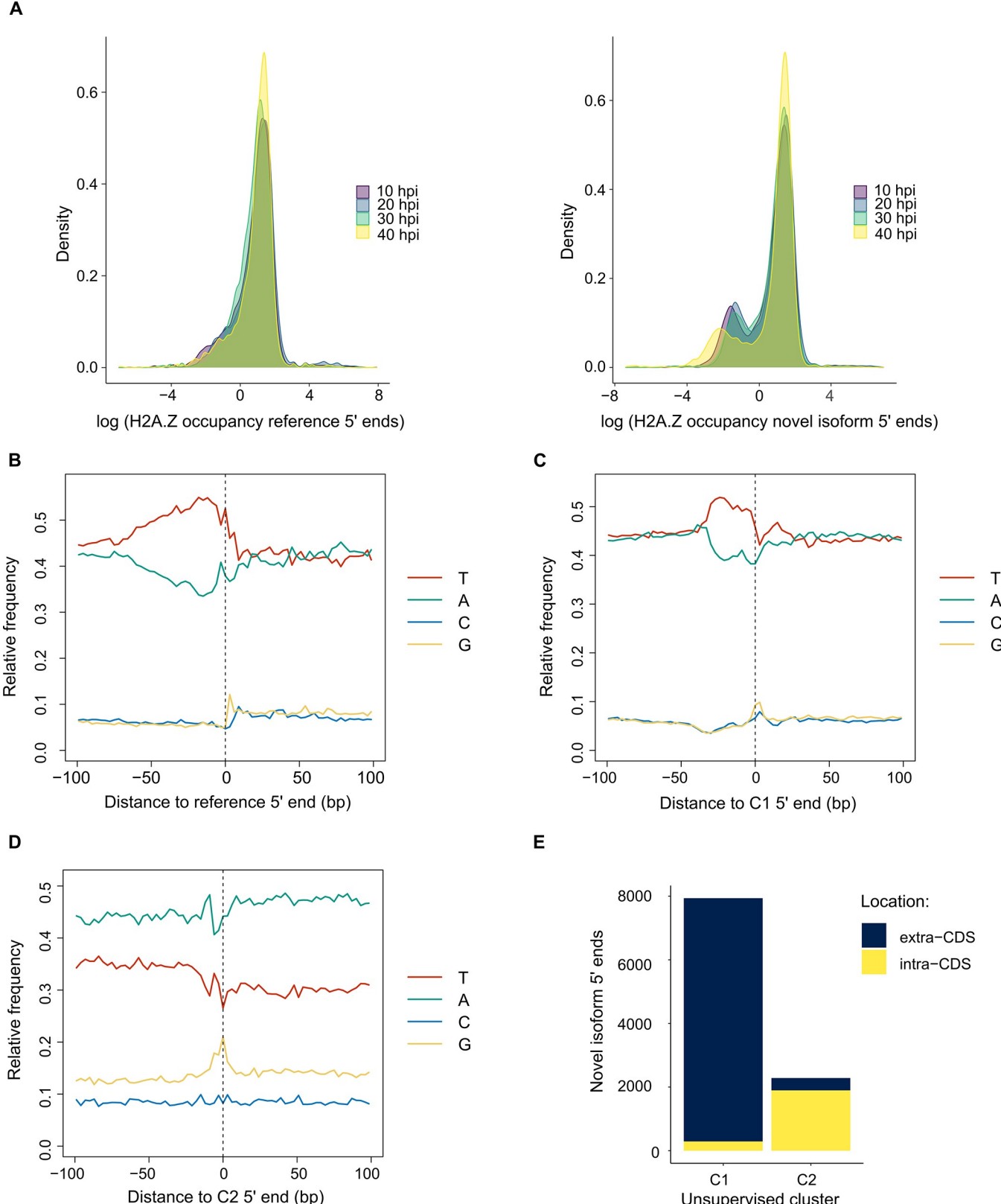

**Fig 3. Analysis of transcript 5′ ends.** Genomic analysis of unique 5′ ends for *Plasmodium falciparum* 3D7 reference transcripts (5,561 positions) and novel isoforms (10,323 positions). **(A)** Distribution of natural log normalized H2A.Z histone occupancy [32] of reference transcripts (left) and novel isoforms (right)

as shown by kernel density plot, in which the vertical axis is the probability density function of the variable (natural log normalized H2A.Z histone occupancy). Data were obtained from parasite samples 10, 20, 30 and 40 hours post-invasion (hpi). **(B)** Average base composition of genome sequence flanking 100 bp either side of reference transcript 5′ ends. **(C)** Average base composition of genome sequence flanking 100 bp on either side of novel isoform 5′ ends for the cluster C1 assigned from H2A.Z histone occupancy data (7,937 positions). **(D)** Average base composition of genome sequence flanking 100 bp on either side of novel isoform 5′ ends for the cluster C2 assigned from H2A.Z histone occupancy data (2,283 positions). **(E)** Location of C1 and C2 novel isoform 5′ ends with respect to annotated protein-coding sequence (CDS) regions.

"antisense" category showed a significantly greater odds ratio in exclusively ring- and schizont-stage datasets. It should be noted that the antisense isoform category does not include transcripts in which the UTRs extending from annotated protein-coding genes overlap adjacent genes on the other strand.

Next, we sought additional evidence to support the expression across the IDC for RI and antisense transcribed regions of the genome represented by novel isoforms. Independent short-read RNA-Seq data obtained using a directional, amplification-free protocol [15], which were not used to construct the isoform catalog, were analyzed for features of interest (Fig 7A and 7B). The RI and antisense features were assigned by unsupervised clustering of short-read RNA-Seq patterns into three groups, following the assumption of three transcriptional transitions during the IDC [37]. The assigned clusters revealed different patterns across the IDC, with clusters showing predominantly schizont-stage expression (40–48 hours post-invasion [hpi]) apparent for both RI and antisense isoforms. Clusters with predominantly ring-stage (0–16 hpi) and trophozoite-stage (24–32 hpi) expression were observed for antisense and RI isoforms, respectively, and a cluster with ring/trophozoite expression (8–24 hpi) was observed for both types of isoform. Notwithstanding the generally lower expression of antisense novel isoforms compared with their reference sense strand counterparts, some antisense isoforms show short-read RNA-Seq coverage along most of their length supporting the annotation from the long-read transcriptomic catalog (Fig 7C).

## Analysis of alternative ORFs and encoded proteins

The novel isoforms may contain alternative ORFs (altORFs) that code for proteins not present in the current proteome annotation [39]. We translated the novel isoform sequences and analyzed all encoded proteins with protein databases. The query sequences were grouped according to BLASTp [40] results (S5 Table) as follows: 15,328 query sequences showed no significant match (no_hit), 6,713 matched *P. falciparum* proteins (Plasmodium_falciparum), and 424 matched proteins from other taxa (other_taxa). The comparison of protein length distributions by Kolmogorov-Smirnov test showed significant differences among all groups (Fig 8A). In the Plasmodium_falciparum group, 1,734 proteins matched a *P. falciparum* 3D7 reference protein with less than 100% identity, but with a median match length of 109 residues (S5B Table). Discrepancies were apparent among the Plasmodium_falciparum group (S5B Table), including i) short matches to *P. falciparum* proteins by altORFs in genomic regions not annotated as protein-coding, i.e., intergenic (50 altORFs) and antisense (34 altORFs) isoforms, ii) short matches to proteins from other organisms when searching the non-redundant database (604 altORFs), and iii) short matches to a non-reference *P. falciparum* protein, i.e., from a different strain, but no match to a *P. falciparum* 3D7 reference protein (480 altORFs). Five proteins from the other_taxa group (S5C Table) also matched proteins with descriptions in the eggNOG database [41].

Given that most altORF-encoded proteins in antisense and intergenic isoforms do not show any similarity to known proteins, additional evidence is needed to determine whether proteins can be produced from these altORFs. Ribosome profiling (ribo-seq) is an established method for assessing ribosome engagement, and is a proxy for protein synthesis [42]. We

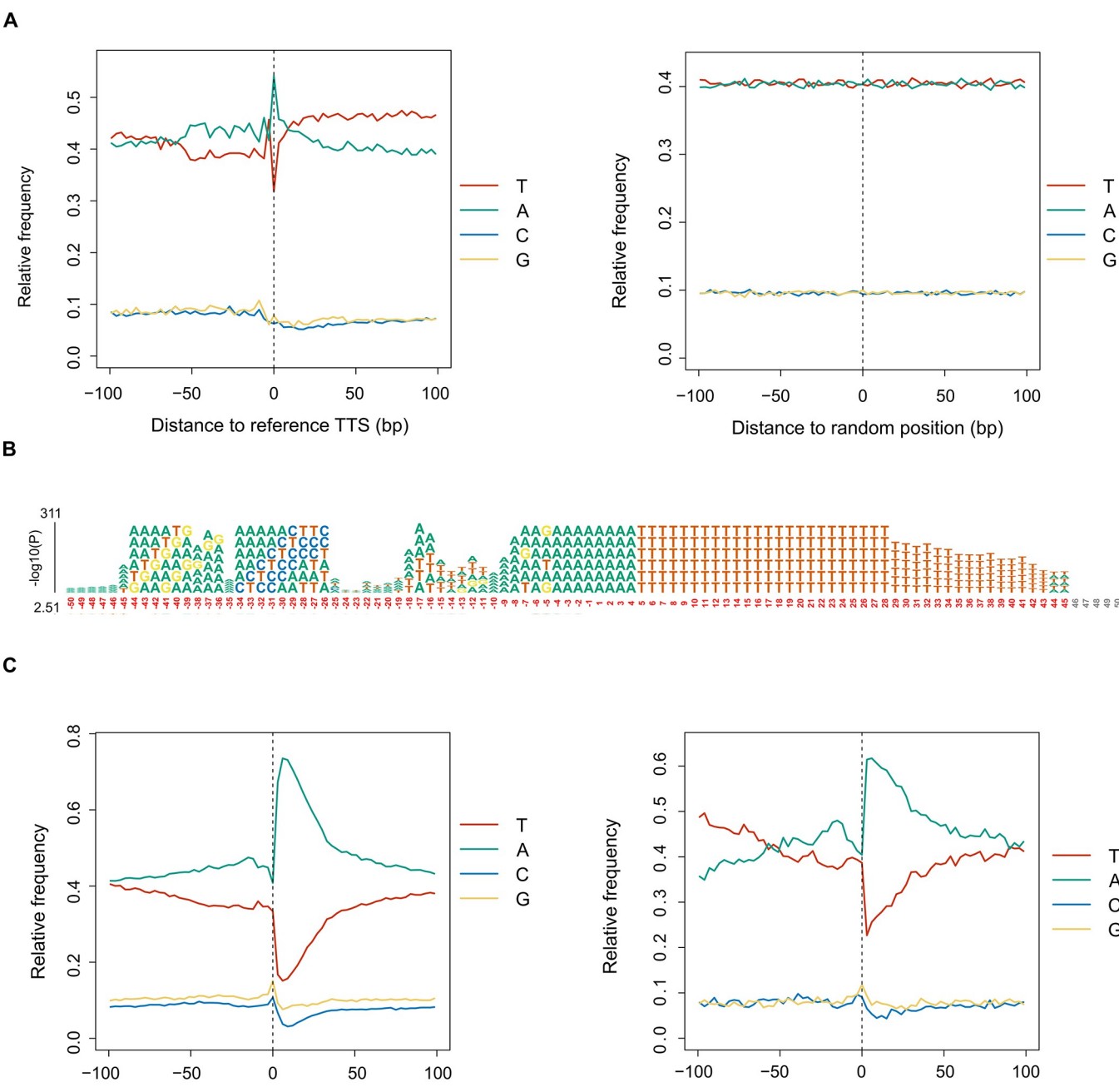

**Fig 4. Analysis of transcript 3′ ends.** Genomic sequence analysis was performed for selected positions from the *Plasmodium falciparum* 3D7 v3.2 reference genome [14]. (**A**) Average base composition of genome sequence flanking 100 bp on either side of reference transcript termination sites (TTS; 5,560 positions) and 5,560 randomly selected positions (right). (**B**) Over-represented 6-mer motifs identified by kpLogo analysis [36] comparing sequences flanking reference TTS with control random positions. Complete tabulated results from kpLogo analysis are shown in S4 Table. (**C**) Average base composition of genome sequence flanking 100 bp on either side of all alternative transcript termination sites (ATTS) for novel isoforms (10,753 positions) and ATTS validated by direct RNA sequencing data (943 positions).

analyzed previously published *P. falciparum* ribo-seq data [10] to assess ribosome engagement in altORFs within antisense and intergenic isoforms, and in ORFs for annotated proteins within reference transcripts. We did not determine ribosome engagement in novel isoforms

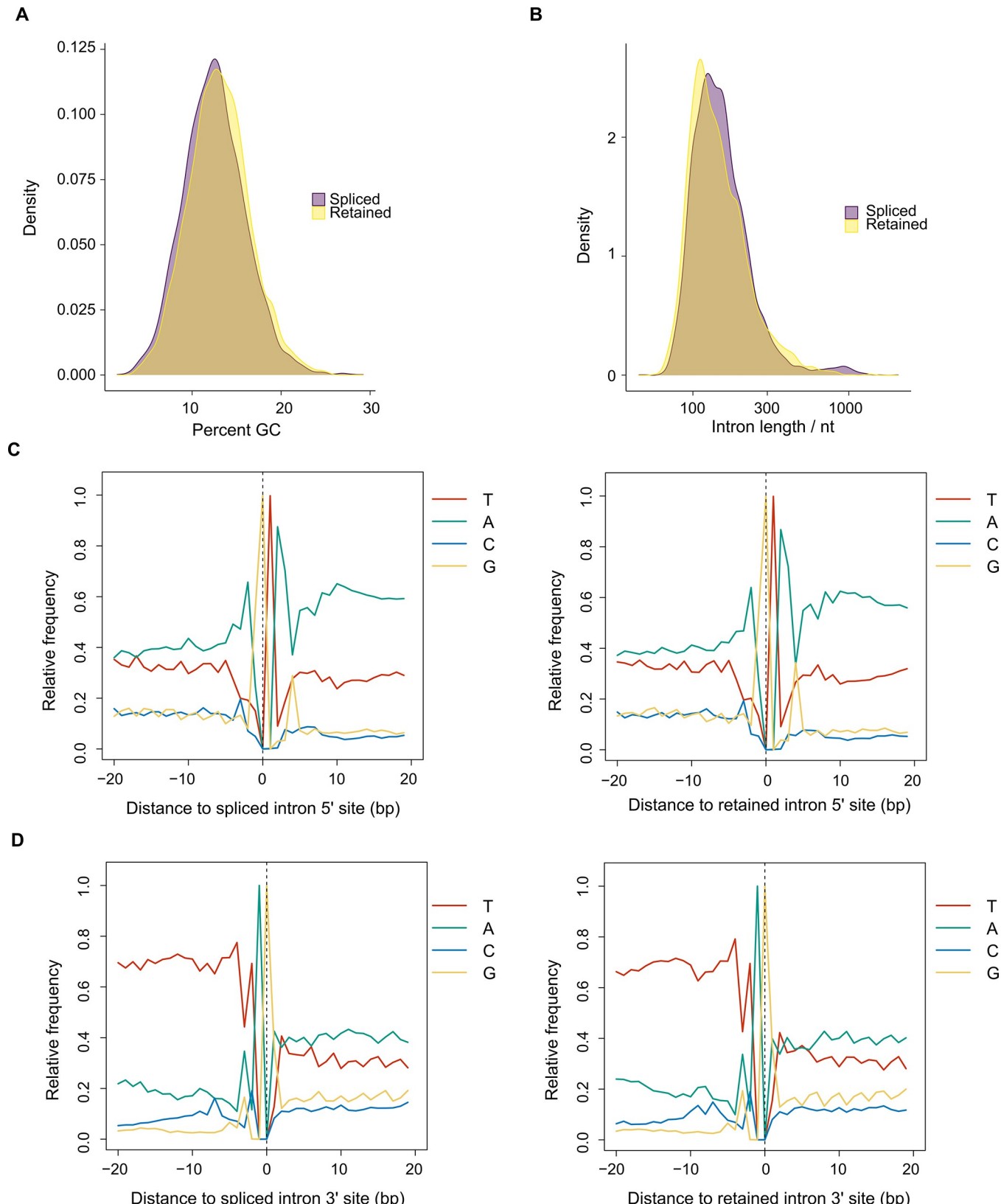

**Fig 5. Analysis of retained introns identified in novel isoforms.** Genomic sequences of retained introns in *Plasmodium falciparum* 3D7 novel isoforms (2,710) were compared with spliced introns (6,258). **(A)** Distribution of intron sequence percent GC content as shown by kernel density plot, in which the

vertical axis is the probability density function of the variable (percent GC). Two-sample Kolmogorov-Smirnov two-sided adjusted $P$ = 3.6e-2 (mean percent GC: retained = 13.08; spliced = 12.51). **(B)** Distribution of intron sequence length as shown by kernel density plot, in which the vertical axis is the probability density function of the variable (intron length). Two-sample Kolmogorov-Smirnov two-sided adjusted $P$ = 1.6e-10 (mean length: retained = 170.98 nt; spliced = 171.21 nt). **(C)** Average base composition of genome sequence flanking 20 bp either side of 5′ splice sites for spliced introns (left) and retained introns (right). **(D)** Average base composition of genome sequence flanking 20 bp either side of 3′ splice sites for spliced introns (left) and retained introns (right).

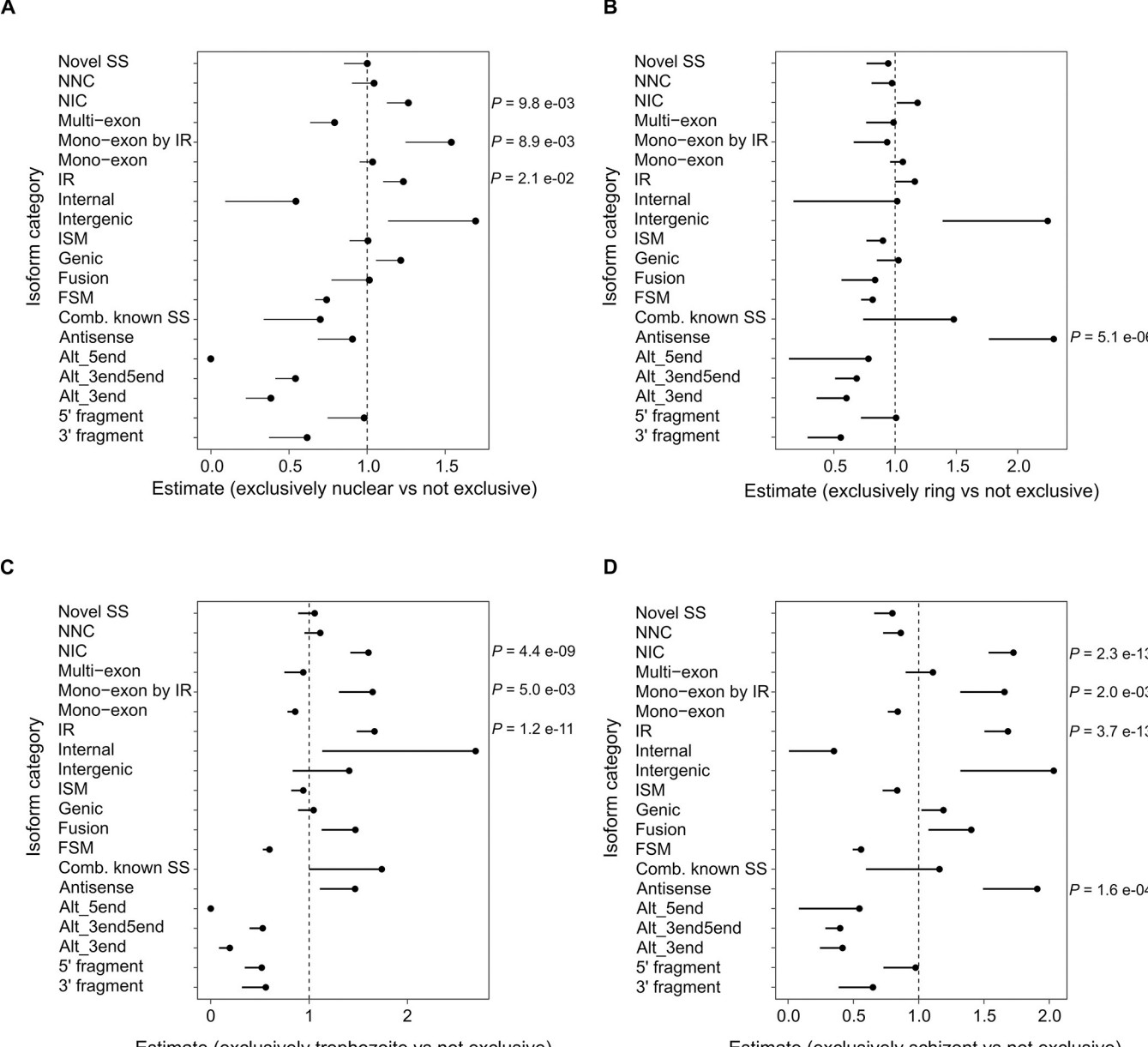

**Fig 6. Over-representation of novel isoform types in the nucleus and across developmental stages.** Counts were made of 11,325 *Plasmodium falciparum* 3D7 novel isoforms assigned to structural categories by SQANTI3 [35] supported by PacBio data in this study (S2B Table). Counts in each category were analyzed by one-tailed Fisher's exact test. Plots of odds ratio for each test are shown; for odds ratios significantly greater than 1 (dashed line), the Holm-Bonferroni adjusted *P*-values are indicated on the right of each plot. The lower confidence interval is indicated for each odds ratio by a solid line. **(A)** Isoforms represented exclusively nuclear vs not exclusive. **(B)** Isoforms represented exclusively ring-stage vs not exclusive. **(C)** Isoforms represented exclusively trophozoite-stage vs not exclusive. **(D)** Isoforms represented exclusively schizont-stage vs not exclusive.

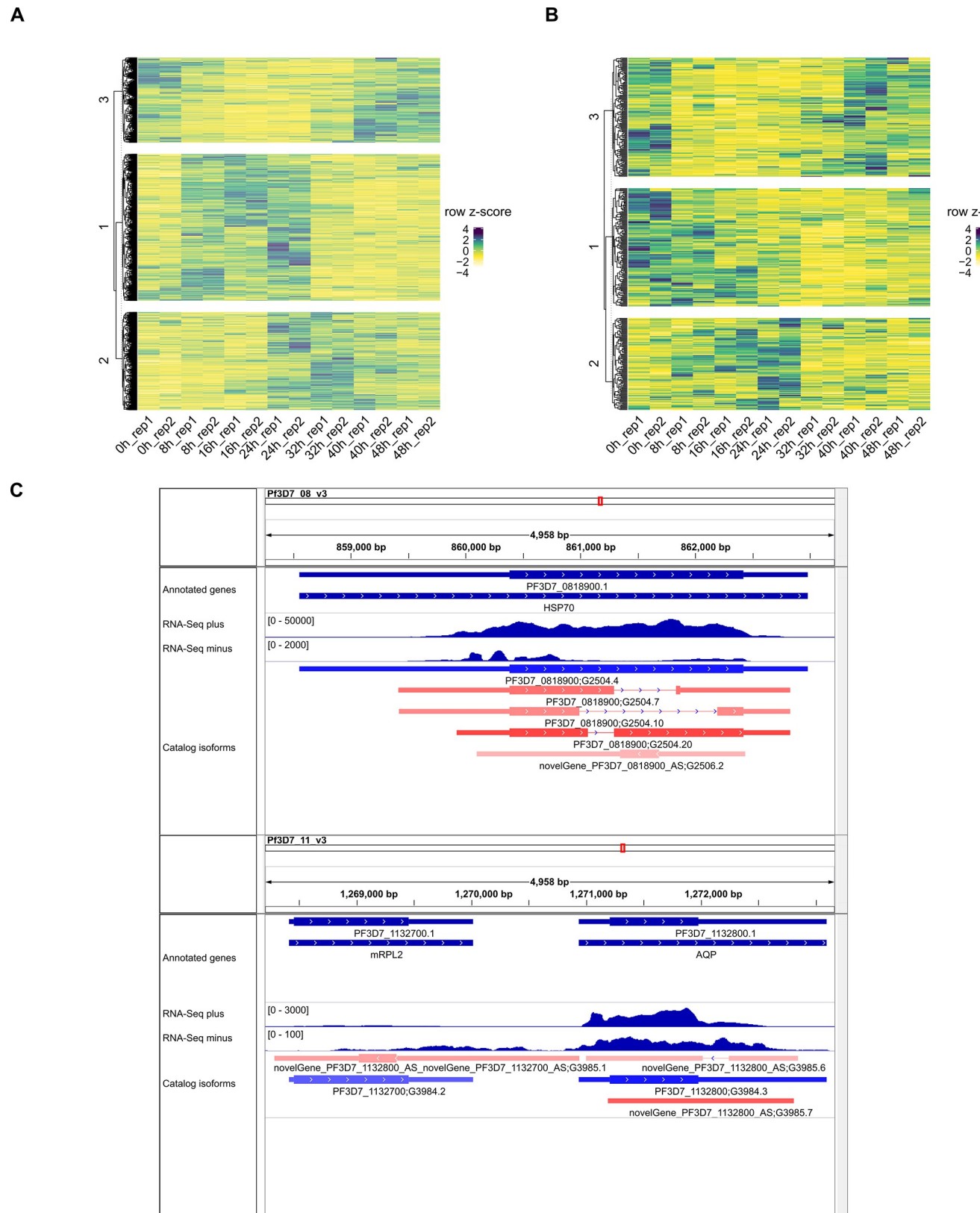

**Fig 7. Short-read RNA-Seq evidence supporting retained introns and antisense isoforms.** Directional, amplification-free short-read RNA-Seq data from *Plasmodium falciparum* 3D7 sampled in duplicate every 8 h post-invasion during the 48 h intra-erythrocytic life cycle [15] were analyzed among regions of interest. **(A)** Heatmap of expression among 2,669 introns retained in novel isoforms. Normalized read counts were row-scaled by z-score, and rows (introns) were k-means clustered (k = 3). The assigned clusters are separated as shown by the dendrogram on the left. 41 rows empty, or with only one value were removed before analysis. **(B)** Heatmap of expression among 415 antisense isoforms. Normalized read counts were row-scaled by z-score and rows (antisense isoforms) were k-means clustered (k = 3). The assigned clusters are separated as shown by the dendrogram on the left. 76 rows empty, or with only one value were removed before analysis. **(C)** Examples of novel antisense isoforms. Screenshots from the IGV genome browser [38] are shown for selected regions of chromosomes 8 (top) and 11 (bottom). The structures of PlasmoDB [16] annotated genes and catalog isoforms are shown in the "Annotated genes" and "Catalog isoforms" tracks, respectively (boxes, exons; thick boxes, open reading frames; lines, introns, and arrows, strand orientation). In the "Catalog isoforms" track, reference transcripts are in blue and novel isoforms are in red. The color hue of each isoform reflects the level of support from long-read RNA-seq data (darker, more support). Antisense isoforms are assigned to "novelGenes". Isoform identifiers (S2 Table) are indicated after the gene identifier. Short-read RNA-Seq coverage (mapped reads) from the merged data from all samples is shown as separate tracks for plus and minus strands.

overlapping annotated genes owing to confounding ribo-seq signals from the reference transcripts. Translational efficiency (TE) and ribosome release scores (RRS) were determined from the data. TE provides a measure of 80S ribosome density normalized to mRNA abundance, which correlates with the abundance of encoded protein [42]. However, TE is not a robust classifier of protein-coding RNA; RRS is a more discriminating metric that distinguishes functional protein-coding transcripts from the pattern of translational termination at the end of the ORF [43]. Ribosome engagement was found to be low overall for altORFs, as TE and RRS could be determined for only 106/417 (25%) of altORFs in intergenic isoforms and 246/1,665 (15%) of altORFs in antisense isoforms (Fig 8B). Moreover, the distributions of TE and RRS in altORFs are significantly different from ORFs in reference transcripts by Kolmogorov-Smirnov test (S6 Table), with the exception of late-trophozoite stage TE for intergenic altORFs. No significant differences in TE or RRS distributions were observed for altORFs in intergenic isoforms compared with antisense isoforms.

In addition to new gene models to accommodate antisense and intergenic isoforms, it may be necessary to accommodate other types of novel isoforms. The transcript catalog includes novel isoforms that overlap two or more genes (S2B Table). We investigated the gene overlaps in more detail and identified 228 novel isoforms that completely span two or three genes among 114 separate genomic regions (S7 Table). Of this group, 168 are polycistronic that span multiple genes in the same orientation (head-to-tail), and 60 span adjacent genes in head-to-head or tail-to-tail opposite orientations. Some polycistronic isoforms code for multiple annotated proteins from different genes, whereas novel proteins are encoded by some polycistronic isoforms owing to isoform events. Finally, we investigated the nearest adjacent annotated features to 167 intergenic isoforms (S8 Table). Eight intergenic isoforms are closest to pseudogenes, seven to telomeres, 12 to annotated non-coding RNA genes, and 140 to protein-coding genes. Gene ontology (GO) analysis of the 140 intergenic isoforms closest to protein-coding genes revealed significant enrichment of the GOslim molecular function term GO:0003700 DNA-binding transcription factor activity (Bonferroni-corrected $P$ = 1.34e-6).

## Discussion

Despite the wealth of RNA-Seq data for *P. falciparum* and the availability of a finished reference genome sequence, transcriptomic annotation is sparse for most genes and does not catalog the full transcriptomic repertoire (isoform diversity) for this species. We addressed this need by creating a transcriptomic catalog from long-read RNA-Seq data. Although long-read RNA-Seq reveals a great diversity of isoforms, technical and biological noise confounds isoform annotation, in particular, the assignment of exon boundaries [27,44]. From the analysis of spike-in RNAs, we observed that filtering of candidate isoforms is required to eliminate 5′ truncated noise (Fig 1A). We used 5′ capped nucleotide signals generated from short-read

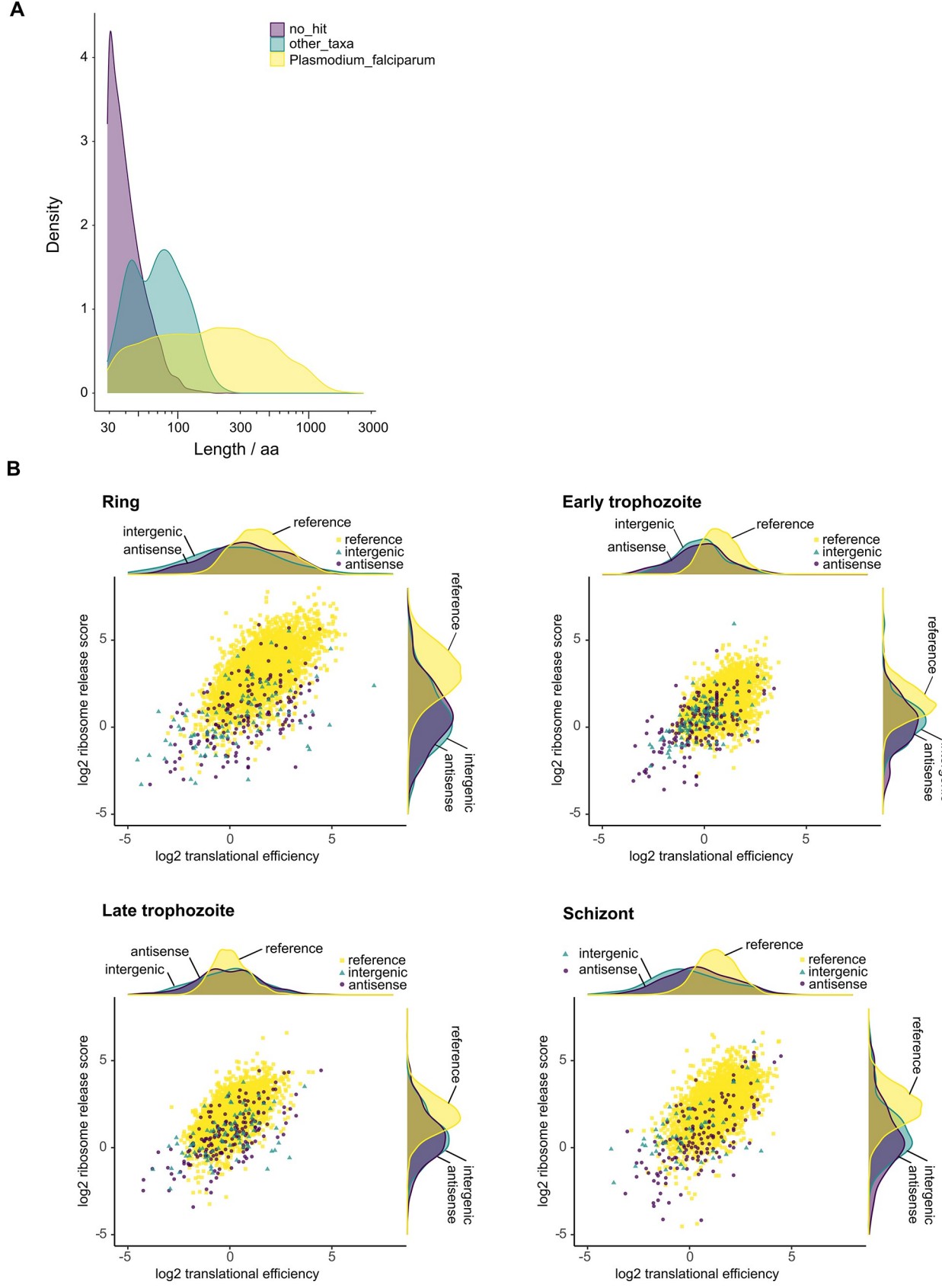

**Fig 8. Analysis of alternative open reading frames (altORFs) and encoded proteins in novel isoforms. (A)** Distribution of protein length (number of amino acid residues) for 22,465 non-redundant protein sequences encoded by altORFs among 12,495 *Plasmodium falciparum* 3D7 novel isoforms. The vertical axis is the probability density function of the variable (number of residues). Protein sequences were searched against protein databases with BLASTp [40] and classified according to the search results. The full BLASTp search results are shown in S5 Table. 15,328 query sequences with no significant match (E value > 1-e4) are shown as no_hit (mean length 43 residues). 6,713 query sequences matched *P. falciparum* proteins (Plasmodium_falciparum) with mean length of 269 residues. 424 query sequences matched proteins from other taxa (other_taxa) with mean length of 77 residues. Two-sample Kolmogorov-Smirnov two-sided test comparing protein length distribution: Holm-Bonferroni adjusted $P<6.6e$-16 (for all three pairwise comparisons). **(B)** Ribosome engagement in intergenic and antisense novel isoforms. Translational efficiency and ribosome release score were determined from ribosome profiling data of ring, early trophozoite, late trophozoite and schizont stages of *P. falciparum* W2 [10] for 3,565 annotated protein-coding ORFs in *P. falciparum* reference transcripts, 106 altORFs in intergenic isoforms and 246 altORFs in antisense isoforms. The distributions of $log_2$ translational efficiency and ribosome release score by kernel density plot are shown above the corresponding scatterplot axes. Pairwise comparisons of distribution were conducted by two-sample Kolmogorov-Smirnov two-sided test. *P*-values from test statistics are shown in S6 Table.

RNA-Seq data to filter *P. falciparum* isoforms. The majority of 5′ capped nucleotide signals are broad, with a mean interquartile width >100 bp [18]. We selected candidate isoforms with 5′-ends not further than 50 bp from the dominant 5′ capped nucleotide in each signal. Furthermore, the reference transcript 5′ ends were assigned the highest priority, such that candidate isoforms with nearby (less than 300 bp) downstream 5′ ends assigned from long-read data were filtered as noise in the transcriptomic merge step of catalog construction. Hence, although ATSS is the most common isoform event (Fig 2), our transcriptomic catalog may still underrepresent the diversity of alternative 5′ ends. We filtered candidate isoforms with non-canonical splicing junctions, because *P. falciparum* is thought to possess minimal splicing machinery that is only capable of splicing canonical (GT:AG and GC:AG) junctions [6]. Although 105 reference transcripts contain non-canonical splicing junctions (S2A Table), all but five are pseudogene transcripts. Non-canonical junctions are not spliced in pseudogenes, as shown by a cDNA analysis of the SURFIN4.1 pseudogene transcript [45]. The non-canonical splicing junctions for reference transcripts of protein-coding genes are present in UTRs, which may represent uncorrected RNA-Seq alignment errors in low-complexity sequences.

## Alternative ends represent the majority of isoform events

ATSS and ATTS represent the majority of isoform events among reference transcripts and novel isoforms (Fig 2), which reflect the most frequent isoform events in the human transcriptome [46]. The majority of ATSS represented among novel isoforms correspond to genomic locations with elevated H2A.Z occupancy throughout the IDC (Fig 3A). Elevated H2A.Z occupancy is a TSS marker in a variety of eukaryotes including *P. falciparum*; furthermore, reduced H2A.Z occupancy upstream of *P. falciparum* genes is associated with transcriptional silencing [47]. The minority of alternative 5′ ends assigned to the C2 cluster correspond to genomic locations with reduced H2A.Z occupancy, different base composition, and are mostly within CDS (Fig 3). Notwithstanding the possibility that alternative 5′ ends can arise from transcription initiation in genomic regions with an unfavorable nucleosomal architecture, we consider C2 5′ ends as unlikely to be transcription initiation sites.

Eukaryotic transcripts with alternative 5′ ends mapping to CDS can be generated post-transcriptionally by co-translational cleavage and recapping [33]. Co-translational cleavage is known to occur at ribosome stalling sites, which include codons for charged residues such as lysine [48]. The ends of endonucleolytically-cleaved mRNA fragments in human cells frequently overlap lysine, GAC (aspartate), and GAA (glutamate) codons [49]. As the same codons are significantly over-represented in the vicinity of C2 5′ ends of novel isoforms (S5 and S6 Figs), co-translational cleavage is plausible. *P. falciparum* can translate mRNA with consecutive lysine codons and other features that typically stall ribosomes more efficiently in human cells without triggering mRNA decay [50]. We conjecture that ribosome stalling

triggers co-translational mRNA cleavage in *P. falciparum*, but translation can continue past the stalling sites, possibly even on endonucleolytically-cleaved mRNA fragments that persist owing to inefficient mRNA decay. The uncapped 5′ ends of endonucleolytically-cleaved mRNA may be recapped via a cytoplasmic mRNA capping complex [33] and manifest as C2 5′ capped ends.

Several different 6-mer motifs are over-represented upstream of reference TTS (Fig 4); however, no single polyadenylation motif was apparent, unlike the majority of metazoan transcripts that possess a conserved upstream AATAAA polyadenylation signal motif [51]. The transcription termination signals in *P. falciparum* are thus similar to other Apicomplexa species that also possess prominent adenine-rich regions in the vicinity of the TTS, but otherwise lack distinct polyadenylation signals [52]. Although similar termination signals were observed among ATTS, we cannot exclude the possibility that ATTS are technical artifacts arising from internally-primed cDNA synthesis (especially those without direct RNA support). Nonetheless, less than 2% of ATTS determined from PacBio data are artifactual because of internal priming [44]. Several novel isoforms terminate within CDS, such that they contain ORFs but lack a stop codon. The eukaryotic non-stop RNA decay mechanism recognizes such mRNAs as aberrant and destroys them [53]; however, the machinery for removing aberrant mRNAs in *P. falciparum* is inefficient because it lacks key non-stop and no-go decay components [50,54]. Novel isoforms lacking a stop codon may be translated if the downstream poly(A) tail functions as a poly-lysine stop to terminate translation. In support of this conjecture, it has been reported that 24 consecutive lysine codons can act as an efficient translation termination signal without triggering mRNA decay in *P. falciparum* [55].

## Retained introns are common isoform events

RI was identified as a frequent isoform event among the novel isoforms (Fig 2). The RI sequences tend to be shorter and possess greater GC content than spliced introns (Fig 5), in common with RI in other eukaryotes [56]. However, the small differences in intron composition and the conservation of splicing junction sequence suggest that other factors are involved in *P. falciparum* RI. From the isoform representation among our PacBio datasets, novel isoforms with RI were more prevalent in nuclear RNA and later (trophozoite and schizont) stages of development (Fig 6). RI clusters with different expression patterns throughout the IDC were resolved using independent short-read RNA-Seq data (Fig 7). A possible connection between nuclear localization and differential patterns of RI across development is the PfSR1 RNA binding protein, which is known to promote splicing in *P. falciparum* [57]. Interestingly, the localization of this protein changes throughout the IDC from predominantly nuclear (ring) to cytoplasmic (schizont) [57–59], suggesting that the reduction of nuclear-localized PfSR1 during IDC progression could generally lead to increased RI and nuclear retention of incompletely spliced isoforms. However, we recognize that it is difficult to make generalizations as RI is likely to be controlled by other factors, including other RNA binding proteins also present in the nucleus, such as PfCELF1 [60].

## Protein-coding potential of altORFs

The majority of altORF-encoded proteins show no similarity to annotated proteins and are also shorter than the 100 residue threshold commonly employed for the annotation of protein-coding genes [61] (Fig 8A); hence, additional evidence is needed to distinguish altORFs encoding small novel proteins from ORFs occurring at random. The altORFs encoding small proteins with discrepant BLASTp hits to *P. falciparum* proteins and hits to proteins in organisms other than *P. falciparum* (S5 Table) are also unlikely to produce proteins. Ribosome

engagement was found to be significantly lower for altORFs in antisense and intergenic iso-forms compared with annotated protein-coding ORFs in the reference transcripts (Fig 8B; S6 Table) indicating that these isoforms represent lncRNA. Although most altORFs may not pro-duce proteins, they may still engage with the translational machinery as regulators. Upstream ORFs (uORFs) are present in the 5′-UTRs of *P. falciparum* mRNA and they act to tune the translational efficiencies of the downstream protein-coding ORF [62–64]. AltORFs, including some that overlap annotated CDS, may act like uORFs to modulate protein synthesis in novel isoforms. Notwithstanding the majority of altORFs that are not likely to produce proteins, some altORF-encoded proteins with long (>100 residues) matches to *P. falciparum* 3D7 refer-ence proteins but with less than 100% identity could represent protein isoforms (S5B Table). *P. falciparum* protein isoforms have been previously reported, including truncated isoforms resolved by two-dimensional differential gel electrophoresis [65]. Some of these truncated iso-forms may be produced by altORFs in novel mRNA isoforms; examples of three genes with reported protein isoforms [65] and expressed as mRNA isoforms with altORFs are shown in S7 Fig.

## Polycistronic, antisense and lncRNA

The annotation of 168 polycistronic novel isoforms that span two or more genes (S7 Table) is in accordance with previous reports of polycistronic mRNA in *P. falciparum* [11,12]. As trans-lation initiation in *P. falciparum* follows the 5′ cap-dependent eukaryotic paradigm [63,64], downstream ORFs in polycistronic mRNA are likely to be translated by less efficient leaky scanning or reinitiation mechanisms [62], or be dependent on specific translational activators such as *var2csa* [66]. The novel isoforms that span adjacent genes on opposite strands include genes previously reported with antisense RNA expression, e.g. *msp2* [67]. The 491 novel iso-forms in the antisense structural category are assigned to novel genes (S2B Table) and thus are distinct from polycistronic isoforms with antisense overlapping regions. The novel isoforms in the antisense structural category thus expand precedent reports of antisense lncRNAs with transcriptional repression [68] and activation functions [69]. The expression of the majority of the antisense structural category novel isoforms is more apparent in ring and schizont stages of the IDC than the trophozoite stage (Figs 6 and 7). Recently, it has been shown that a greater proportion of protein-coding mRNA is present in RNA duplexes in ring and schizont stages compared with the trophozoite stage, and that sense-antisense transcript pairs can form stable RNA duplexes [70]. Increased stability of mRNA when duplexed with an antisense partner could explain the general trend of increased mRNA stability in the schizont stage [71,72], per-haps leading to the storage of some transcripts that persist until the early ring stage after rein-vasion. In addition to antisense novel isoforms, 167 intergenic isoforms were annotated in the catalog. From nearest-feature analysis (S8 Table), the intergenic isoforms near telomeres are consistent with a previous report of lncRNAs transcribed from *P. falciparum* subtelomeric regions [73]. The enrichment of the GO term DNA-binding transcription factor activity among genes near intergenic isoforms could indicate that these intergenic RNAs have impor-tant regulatory functions in shaping the local chromatin environment, e.g., by recruiting chro-matin modifiers that permit control of transcription in response to stimuli [74].

## Limitations of the study

The transcriptomic catalog presented in this study does not include isoforms expressed in other stages of development, i.e., gametocyte, mosquito, and liver stages. With improvements in long-read RNA-Seq technologies (particularly minimum sample requirements), these stages will become accessible to transcriptomic annotation in the future. We did not quantify isoform

abundance across the IDC from the stage-specific PacBio library data because quantification accuracy is limited by the low depth of sequencing compared with short-read RNA-Seq, and the non-uniform coverage of isoforms in which 5′ ends are less likely to be covered by sequencing reads compared with 3′ ends [75]. Isoform abundance across the IDC can be quantified using alternative methods such as isoform-specific reverse-transcription quantitative PCR [76]. The assessment of altORF protein-coding potential from ribo-seq data was limited to intergenic and antisense isoforms. The altORFs in other isoforms can be assessed using alternative approaches such as global translation initiation sequencing [77] and N-terminal proteomics [78].

In conclusion, the *P. falciparum* transcriptomic catalog described herein adds an important resource for this pathogen. The details of isoform structures, classification, and isoform events are provided for future studies (S1 Dataset, S2 and S3 Tables). The major isoform events highlighted in this work (ATSS, ATTS, and RI) prompt questions to be addressed, including what mechanisms are responsible for generating these isoform patterns, and what biological roles (if any) these isoforms play? The encoded proteins from the novel isoforms (S5 Table) can be used to test whether altORFs contribute toward an expanded proteome, with the caveat that validation is challenging with current proteomic technologies [79].

## Materials and methods

### Ethics statement

Human erythrocytes and serum were obtained from donors after providing informed written consent, following a protocol approved by the Ethics Committee, National Science and Technology Development Agency, Pathum Thani, Thailand, document no. 0021/2560.

### In vitro culture of *Plasmodium falciparum*, subcellular fractionation, and RNA extraction

*Plasmodium falciparum* reference strain 3D7 (NCBI: txid36329) was used for all transcriptomic experiments. To assess the method for isolation of nuclei, transgenic *P. falciparum* GAL4-GFP expressing a nuclear-localized GFP reporter protein [80] was used. Parasites were cultured in vitro in human O+ erythrocytes and buffered Roswell Park Memorial Institute 1640 medium containing 8% pooled, heat-inactivated human serum as described previously [23]. Cultured parasites were synchronized by 60% percoll gradient separation to enrich late stages followed by enrichment of ring stages by sorbitol treatment [81]. Parasites were harvested immediately (ring stage), 12 h (trophozoite stage), and 24 h (schizont stage) after synchronization. Parasites were liberated from host cells by treatment with 0.1% (w/v) saponin. Parasite pellets were washed three times with phosphate-buffered saline (1x PBS; 137 mM NaCl, 2.7 mM KCl, 8 mM $Na_2HPO_4$, and 2 mM $KH_2PO_4$). For extraction of total RNA, the washed pellet of saponin-liberated parasites was suspended in TRIzol™ reagent and RNA was extracted according to the manufacturer's instructions (#15596026, Invitrogen). For subcellular fractionation, saponin-liberated parasites (obtained from independent synchronized cultures from those used to prepare total RNA) were disrupted by trituration in 0.3 mL of parasite lysis buffer (20 mM HEPES, pH 7.8, 10 mM KCl, 15 mM $MgCl_2$ 1 mM DTT, 1x cOmplete™ protease inhibitor cocktail no EDTA (#04693132001, Roche), 40 U/mL SUPERas•In RNase inhibitor (#AM2694, Invitrogen), and 0.65% Nonidet P-40). Nuclei were pelleted by centrifugation at 2500 × g for 5 min and the supernatant was removed by aspiration. Nuclei were washed three times with 1 mL parasite lysis buffer. Nuclei were resuspended in glycerol containing Hoechst 33342 for microscopic analysis. Nuclei were mounted on a glass slide and

visualized using an Olympus FV1000 confocal microscope. The Hoechst 33342 signal was obtained using 346/460 (excitation/emission) filters, and the GFP signal was obtained using 490/525 (excitation/emission) filters. RNA was extracted from nuclei using TRIzol™ reagent, as described for total RNA. Purified RNA samples were stored in 75% ethanol at -80˚C before use. On the day of cDNA synthesis, the RNA was dried and re-suspended in nuclease-free water. RNA concentration was estimated by NanoDrop ND-1000 (Thermo Scientific), assuming $A_{260}$ = 1.0 is equivalent to 40 μg/mL RNA.

## Production of recombinant HseIF4E-eIF4G_x6His fusion protein

Human eIF4E and eIF4G1 gene sequences were obtained by reverse-transcription PCR using Universal Human Reference RNA as a template (#740000, Agilent). The RNA was reverse transcribed with oligo dT and Superscript IV enzyme as recommended by the manufacturer (#18090010, Invitrogen). The eIF4E and eIF4G1 gene sequences were PCR-amplified with Phusion High-Fidelity DNA polymerase (#F-530XL, Thermo Scientific) and primer pairs Hs4EF/Hs4ER and Hs4G1F/Hs4G1R (S9 Table), respectively. The amplified eIF4E and eIF4G gene sequences were ligated via a common KpnI restriction site to make a fusion gene, as previously described [24]. The fusion gene (1.1 kb) encodes a fusion protein with eIF4E and eIF4G1 domains separated by a flexible linker peptide (GGGGSGGGGTGGGGS) and a C-terminal six-histidine tag to allow immobilization of the fusion protein on a $Ni^{2+}$ solid matrix. The fusion gene PCR product was cloned into the pET17b vector (Novagen) via the unique EcoRI and XhoI sites and sequenced by the Sanger dideoxy method (Macrogen). The recombinant plasmid, named HseIF4E_eIF4G_x6His (S1 Text), was transformed into *Escherichia coli* strain BL21(DE3) for expression of recombinant protein. The 5′ cap-binding activity of the recombinant protein was assessed by specific elution from a 5′ cap analog matrix as previously described [82].

## Purification of HseIF4E-eIF4G fusion protein and 5′ capped RNA

A 1 L culture of *E. coli* BL21(DE3) cells transformed with HseIF4E-eIF4G_x6His was harvested after 20 h induction with 0.4 mM isopropyl ß-D-1-thiogalactopyranoside at 20˚C. The induced cells were re-suspended in 30 mL of ice-cold buffer Ni-A (300 mM NaCl, 50 mM imidazole, 50 mM $Na_2HPO_4$, and 15% glycerol pH 8.0) and disrupted by French Press. The disrupted cell suspension was clarified by centrifugation and filtration and applied to a 5 mL HisTrap FF $Ni^{2+}$-Sepharose column (GE Healthcare) equilibrated with buffer Ni-A at a flow rate of 1 mL/min controlled using an ÄKTA system (GE Healthcare). After sample loading, the column was washed with buffer Ni-A until the UV signal reached the baseline. The recombinant protein was eluted in a 200 mL gradient of 50% buffer Ni-B (300 mM NaCl, 500 mM imidazole, 50 mM $Na_2HPO_4$, and 15% glycerol pH 8.0). Peak elution fractions were pooled and diluted ten-fold in buffer Q-A (50 mM Tris pH 8.0, 50 mM NaCl, and 15% glycerol). Diluted protein was applied to a 10 mL Q-Sepharose FF column (GE Healthcare) equilibrated in buffer Q-A at a flow rate of 1 mL/min controlled using an ÄKTA system. After sample loading, the column was washed with buffer Q-A until the UV signal reached the baseline. Recombinant HseIF4E-eIF4G_x6His protein was eluted in a 200 mL gradient of 50% buffer Q-B (50 mM Tris pH 8.0, 1000 mM NaCl, and 15% glycerol). Peak fractions of eluted protein were combined, aliquoted, and stored at -80˚C before use. Protein concentration was determined by Bradford assay (Biorad).

In Nanopore RNA-Seq experiments, RNA sequin synthetic spike-in RNAs [26] were added to parasite RNA to guide downstream data analysis. MixA RNA sequins were a gift from Dr. Jim Blackburn (Garvan Institute, Sydney, Australia) and were 5′ capped using the Vaccinia

virus 5′ capping system, following the manufacturer's recommended protocol (New England Biolabs). 10–30 ng of the 5′ capped RNA sequin mixture was added to *P. falciparum* total RNA (extracted from mixed IDC stages) before purification of 5′ capped RNA.

For purification of 5′ capped RNA, purified HseIF4E-eIF4G_x6His protein (50 μg) was immobilized on 40 μL of HisPur Ni-NTA Magnetic Beads (#88831, Thermo Scientific) equilibrated with RNase-free 1XPBS (Ambion). A 50–100 μg sample of *P. falciparum* total RNA (with or without spike-in RNA) was incubated with protein-immobilized beads to capture 5′ capped RNA. Beads were washed three times with 1 mL 1XPBS, and 5′ capped RNA was eluted in a solution of 1% SDS, 200 mM NaCl, and 1 mM EDTA. 5′ capped RNA was purified by acidic phenol:chloroform extraction and ethanol precipitation. The 5′ capped RNA pellet was re-dissolved in 20 μL of nuclease-free water, and genomic DNA was removed using a TURBO DNA-free kit, following the manufacturer's instructions (#AM1907, Invitrogen).

## PacBio Sequel RNA-Seq

Samples of purified 5′ cap-enriched RNA (100–300 ng) were converted to cDNA using a SMARTer™ PCR cDNA Synthesis Kit (#634928, Takara), as recommended by the manufacturer. Double-stranded cDNA was PCR-amplified following the recommendations for PacBio Iso-Seq Template Preparation for Sequel Systems (PacBio). The amplified DNA was initially purified using 0.4× AMPure beads, and a portion of each was gel-purified to enrich for fragments >4 kb using a Gel extraction kit (Qiagen). The gel-purified fraction was combined with the remainder of non-size selected DNA. The pooled DNA was purified using 0.4× AMPure beads. Purified DNA samples were sent on dry ice to Novogene AIT (Singapore) for PacBio Sequel sequencing service. PacBio Sequel raw data were delivered in.bam format. The raw data were processed using the Iso-Seq 3 application in the SMRT Link v6.0 program suite (PacBio). Full-length, non-chimeric circular consensus sequences (FLNC CCS) were reported from Iso-Seq3. The term "full-length" refers to sequences with cDNA adapters at both ends, although incomplete cDNA that does not extend to the original 5′ end of the mRNA can acquire a 5′ adapter by template switching during cDNA synthesis [83]. Low quality (lq, accuracy < 0.99) and high quality (hq, accuracy >0.99) sequences were reported in.fastq format. The lq and hq sequences were aligned separately to the *P. falciparum* 3D7 v3.2 reference genome [14] using the minimap2 tool [84] in the spliced alignment mode guided by PlasmoDB-53 reference genome annotation with maximum intron 5 kb, settings: -ax splice:hq -uf -G 5k —secondary = no—junc-bed. Reference genome sequence and annotation files were downloaded from PlasmoDB [16].

Candidate non-redundant isoforms were generated from the aligned reads using the Transcriptome Annotation by Modular Algorithms (TAMA) program suite [27]. Aligned read.sam outputs from minimap2 were used as inputs for the TAMA collapse tool with "low" [27] settings: -x capped -rm low_mem -sj sj_priority -log log_off -lde 5 -sjt 20 -a 100 -z100 -m 20.

## Nanopore RNA-Seq

First-strand cDNA synthesis reactions contained purified 5′ capped RNA spiked with RNA sequin (100–300 ng) as described above, RTSMARTCDS primer (S9 Table), and Superscript IV (Thermo Scientific). RNA was removed by treatment with 0.2 N sodium hydroxide, and first-strand cDNA was purified using 0.5× AMPure XP beads as recommended by the manufacturer (#A63880, Beckman Coulter). A double-stranded 5′ adapter was made by annealing SPLADAP and SPN6 oligonucleotides (S9 Table). 5′ adapter was ligated to 50–100 ng cDNA using T4 DNA ligase (New England Biolabs). The excess adapter was removed, and cDNA purified using 0.5× AMPure XP beads. Double-stranded cDNA was obtained by PCR

amplification with primers ABRIDGEDPE1 and SMARTPCRII (S9 Table) and KAPA HiFi HotStart Ready mix (Kapa Bioscience). The reaction conditions used were: 95˚C 5 min; 15–20 cycles of 98˚C 15 s, 62˚C 5 min 30 s. DNA was purified using 0.5× AMPure XP beads and converted to a Nanopore DNA sequencing library using a 1D ligation kit as recommended by the manufacturer (#SQK-LSK109, Oxford Nanopore Technologies). The DNA sequencing library was quantified by Quant-iT dsDNA HS assay (#Q33120, Invitrogen) and agarose gel electrophoresis. Approximately 20 fmol of library DNA was sequenced on a SpotON MkI flow cell (R9.4) in a Minion device (Oxford Nanopore Technologies). Data were collected over 24 h runs using MinKnow software (Oxford Nanopore Technologies). The signals were converted to sequences in.fastq format using Scrappie basecalling software (Oxford Nanopore Technologies). Cutadapt version 2.8 [85] was used to identify reads containing both 5′ and 3′ cDNA adapters, trim adapters, and reverse-complement first-strand cDNA reads. The settings used were: cutadapt—rc—discard-untrimmed -g CTACACGACGCTCTTCCGATCT...AAAAAAAA AGTACTCTGCGTTGATACCACTGCTT -e 0.2. The processed reads were aligned to the *P. falciparum* 3D7 v3.2 reference genome using the minimap2 tool in the spliced alignment mode guided by genome annotation with maximum intron 5 kb with settings: minimap2 -ax splice -uf -k14 -G 5k —secondary = no—junc-bed. For assessing the accuracy of RNA sequin isoform annotation, alignment was performed to the combined *P. falciparum* 3D7 v3.2/ RNA sequin in silico chromosome (chrIS) reference genome, guided by the combined reference genome annotation with the same parameters, except that the maximum intron was increased to 550 kb (-G 550k) to account for the larger introns in the chrIS gene models. Candidate non-redundant isoform sequences were generated from reads aligned to chrIS using different tools. The TAMA collapse tool was employed with "low" settings: -x capped -rm low_mem -sj sj_priority -log log_off -lde 5 -sjt 20 -a 100 -z100 -m 20 -icm ident_map and "high" settings: -x capped -rm low_mem -sj sj_priority -log log_off -lde 1 -sjt 20 -a 300 -z300 -m 20 -icm ident_map. The StringTie2 tool [28] was employed with the following settings: long read (-L), no assembly (-R), and disable trimming (-t). The FLAIR tool [29] was employed guided by the chrIS annotation. Candidate non-redundant isoforms were called by the FLAIR tool using the flair.py script with the following settings:—no_end_adjustment—trust_ends -n longest -s 2. The consistent and inconsistent isoforms returned by FLAIR were combined for further analysis. Candidate non-redundant isoform sequences reported by each tool were compared with the chrIS annotation using the GffCompare tool [30].

## Generation of candidate non-redundant isoforms from external long-read RNA-Seq data

Direct RNA sequencing data obtained from the Nanopore platform [22] were downloaded from the Sequence Read Archive database under accession number SRR11094274. The raw data were processed using the isONcorrect pipeline [86]. The processed reads were aligned to the *P. falciparum* 3D7 v3.2 reference genome, guided by the PlasmoDB-53 reference genome annotation using the minimap2 tool as described above for Nanopore RNA-Seq. Candidate non-redundant isoforms were obtained from the aligned read data using TAMA collapse low as described above.

Data from two cDNA libraries sequenced using PacBio Sequel [21] were downloaded as raw.bam files from the Genome Sequence Archive (accession numbers CRR216516– CRR216520). FLNC CCS sequences were obtained from the data as described above for PacBio Sequel RNA-Seq. The high-quality and low-quality FLNC CCS from each of the two libraries constructed were aligned separately to the *P. falciparum* 3D7 v3.2 reference genome using the minimap2 tool and candidate non-redundant isoforms obtained using TAMA collapse low as

described above for PacBio Sequel RNA-Seq. RNA-Seq data obtained using PacBio RS [15] were downloaded from the European Nucleotide Archive (accession numbers ERR2282005– ERR2282008). FLNC CCS were obtained using SMRT analysis 2.3.0 (PacBio). The FLNC CCS were aligned to the *P. falciparum* 3D7 v3.2 reference genome using the minimap2 tool and candidate non-redundant isoforms were obtained using TAMA collapse low as described above for PacBio Sequel RNA-Seq. Because only 31 isoforms were generated by TAMA collapse low (S1 Table), we decided not to use these data for constructing the transcriptomic catalog.

### Analysis of external short-read RNA-Seq data

Directional, amplification-free short-read RNA-Seq data for *P. falciparum* 3D7 strain parasites sampled at 8 h intervals during the IDC [15] were downloaded from the European Nucleotide Archive database (accession numbers ERR2234550–ERR2234556 and ERR2234564– ERR2234570). The data were processed with fastp [87] using default parameters and aligned to the *P. falciparum* 3D7 v3.2 reference genome using the HISAT2 tool [88] guided by the reference genome annotation with maximum intron 5 kb. Simplified annotation files were created for retained intron and antisense features and used with the aligned read.bam files to obtain counts of mapped reads with the featureCounts tool [89] with options selected for paired, strand-specific reads (-p -s 2). The raw counts for each feature were normalized to the total of mapped reads in each dataset and scaled to counts per million mapped reads (cpm). Heatmaps were generated using the ComplexHeatmap package in R [90]. Data frames of cpm values for features of interest were used as input to ComplexHeatmap, in which cpm values were transformed to row z-scores and features (rows) were clustered using the k-means algorithm with k = 3. RNA-Seq coverage tracks for visualization in the IGV tool [38] were created from the. bam files. The.bam files were filtered to retain the first read from each pair using the SAMtools program [91] view function, with the following options: -b -F 64. Bedgraph coverage files for each strand were generated from the filtered.bam files using BEDTools [92], which were then used to generate BigWig files using bedGraphToBigWig [93].

### Annotation of 5′ capped nucleotides for validating 5′ ends of novel isoforms

5′ capped nucleotide signals were obtained from CAPture-enrichment [18] and 5′ cap end sequencing [17] experimental data reported previously. The signals were annotated from these data using the CAGEr program [94] as previously reported [18]. The signals (CAGEr clusters) were concatenated, i.e., the union of signals from different data sources, into a single "CAGE peak BED file" for SQANTI3 [35]. The "dominant.ctss" nucleotide reported by CAGEr for each signal was assigned as the middle point of each signal, as per the requirement for SQANTI3.

### Construction of transcriptomic catalog

Candidate non-redundant isoforms in.bed12 format obtained from TAMA collapse analysis of each long-read sequencing dataset were merged with the PlasmoDB-53 reference genome annotation using TAMA merge [27] with the following parameter settings: -a 300 -z 300 -m 20 -d merge_dup. Under these settings, isoforms with 5′ or 3′ ends within 300 bp of the common end for the group were collapsed. TAMA merge was run with the reference transcripts set to cap-mode and full priority (1,1,1), and all test data candidate isoforms were set to no-cap mode. Candidate isoforms from PacBio hq FL CCS data were given the next highest priority (2,2,2), followed by PacBio lq FL CCS and Nanopore cDNA (3,3,3). Direct RNA isoforms were given second-highest priority for transcript 3′ ends (3,3,2). The isoforms from TAMA merge

were used as input together with the CAGE peak BED file described above and the PlasmoDB-53 reference annotation to the SQANTI3 tool for classifying isoforms [35]. The output_classification.txt file from SQANTI3 and the tama_merge_trans.txt file from TAMA merge were used for transcript filtering. Reference transcripts were defined as isoforms classified by SQANTI3 under the "structural_category" as "full-splice_match" if both "diff_to_TSS" and "diff_to_TTS" were less than 50 bp, i.e., the 5′ and 3′ ends were within 50 bp of reference transcription start site (TSS) and transcription termination site (TTS) (S2A Table). All putatively novel isoforms not classified as reference transcripts were filtered with the following criteria: (i) length greater than 200 bp, (ii) all-canonical splice junctions, (iii) "dist_to_cage_peak" less than 50 bp (evidence of 5 capped end), and (iv) candidate isoform detected in more than one biological sample (using the tama_merge_trans.txt output from TAMA merge for guidance). 12,495 novel isoforms remained after filtering (S2B Table).

## Characterization of isoform events

The structural categories and subcategories of novel isoforms reported by SQANTI3 were used for the initial assessment of isoform events. 11,325 novel isoforms are represented among one or more PacBio datasets generated in this study (S2B Table). Isoform events were investigated among these isoforms as follows. For each SQANTI3 structural category and subcategory, the isoforms represented exclusively among one or more nuclear-sourced libraries (i.e., not represented in total RNA) were tallied. The isoforms represented among one or more total RNA-sourced libraries were tallied for comparison. One-way Fisher's exact tests were performed using the rstatix package in R [95], with the null hypothesis that the odds ratio of exclusively nuclear isoforms was not greater than one. Holm-Bonferroni-corrected $P < .05$ was considered significant. The same analysis was performed for isoforms represented exclusively among ring, trophozoite, and schizont stage-sourced libraries compared with isoforms represented by libraries from two or three different developmental stages (not exclusive to one stage). The odds ratios and associated lower confidence intervals from row-wise Fisher's exact tests were plotted using the ggplot2 package in R [96].

Isoform events were identified among novel isoforms and reference transcripts using the CATANA tool [97] with default settings. The transcript annotation (.gtf) file together with the PlasmoDB-53 annotation were used as inputs for CATANA. Isoform events among the reference transcripts were identified and separated. Isoform events were filtered by retaining events reported as exclusive to the reference transcripts to account for redundancy among novel isoforms. The sequences of 2,710 retained introns (identified as novel events by CATANA) and spliced introns (all introns in the reference transcripts with no associated retention event detected by CATANA; 6,258 sequences) were obtained from the *P. falciparum* 3D7 v3.2 reference genome using BEDTools. Intron sequences shorter than 34 bp were excluded because flanking partial exon sequences could confound pattern analysis. The percent GC and length values for each intron sequence were obtained using the SeqKit tool [98]. The distributions of intron length and percent GC were plotted using the ggplot2 package in R. Kolmogorov-Smirnov two-sided, two-group tests comparing the distributions of retained and spliced introns were performed using the stats package in R [95], in which Holm-Bonferroni adjusted $P < .05$ was considered significant.

Novel isoforms completely spanning two or more annotated protein-coding genes, and the nearest annotated features to intergenic isoforms were identified using BEDTools with intersect and closest functions, respectively. Gene ontology (GO) analysis of 140 intergenic isoforms closest to protein-coding genes was performed using the PlasmoDB GO enrichment analysis web application [16].

## Analysis of transcript 5′ ends

Transcript 5′ end positions were obtained from the.gff annotation files for reference and novel isoform transcripts. H2A.Z ChIP-seq data reported in a previous study [32] were used to calculate genome-wide normalized occupancy as described previously [18]. The normalized occupancy values at the transcript 5′ end positions were natural log-transformed and the distributions were plotted using the ggplot2 package in R. Principal component scores of the log-transformed normalized occupancy values at novel isoform 5′ end positions were calculated using the MacroPCA package in R [99]. The first principal component scores were used for unsupervised clustering of 5′ end positions using the cross entropy clustering (CEC) package in R [100] with card.min = 15%. Under this setting, CEC resolved two clusters, designated as C1 and C2. Plots of average base frequency in the vicinity of 5′ end positions were performed using the seqPattern package in R [101].

Codon composition analysis was performed for C2 cluster positions within annotated protein-coding sequence (CDS) regions (1896 positions). An SQLite database was constructed for these positions, together with a.bed file of CDS region coordinates and a.fasta file of CDS region sequences using an in-house script (available from: https://github.com/PavitaKae/PrepareCodonComposition/blob/main/preparecodoncomposition.py). The SQLite database,.bed file, and.fasta files were provided as input to the codon composition tool [49]. The tool outputs the counts of all 64 codons within 20 bp windows spanning positions of interest and randomly selected positions from the same genes. The distributions of codon counts between C2 and random position groups were compared by two-sided Kolmogorov-Smirnov tests using an in-house script (available from: https://github.com/PavitaKae/PrepareCodonComposition/blob/main/ks.py). Holm-Bonferroni adjusted $P < .05$ was considered significant.

## Analysis of transcript 3′ ends

Transcript 3′ end positions were obtained from the annotation files for reference and novel isoform transcripts. Plots of average base frequency were performed using the seqPattern package in R. 5,560 random positions were obtained from the *P. falciparum* 3D7 v3 reference genome using BEDTools. Genomic sequences flanking random positions (control set) and reference TTS were obtained using the seqPattern package in R. Analysis of over-represented 6-mers in the vicinity of reference TTS was performed using the kpLogo tool [36] using the control set sequences as background and other settings as default.

## Analysis of encoded proteins in novel isoforms

Novel isoform sequences in.fasta format were obtained using GffRead [30]. The sequences were translated using the orfipy tool [102], with all complete ORFs considered and a minimum protein length of 29 amino acid residues (equal to the smallest currently annotated *P. falciparum* 3D7 protein PF3D7_1133000.1). The output of 40,036 was filtered to 22,465 non-redundant protein sequences with the SeqKit tool. Encoded protein sequences were queried using the BLASTp web application hosted by NCBI [40] against the non-redundant and *P. falciparum* 3D7 databases using default parameters. BLASTp hits with E-value <1e-04 were considered significant, but only the top-ranked hits were recorded. Significant hits to *P. falciparum* proteins were considered a greater priority than hits to other species with equal or lower E-values. The queried protein sequences were also submitted to the eggNOG-mapper v2 web application [41] and searched with default parameters.

## Analysis of protein coding function from ribosome profiling

Translation of ORFs in reference transcripts and novel isoforms during the *P. falciparum* IDC was assessed using ribosome profiling (ribo-seq) data published in Caro et al [10]. We first

generated a *P. falciparum* W2 strain variant-corrected genome index for read alignment as this strain was used for ribo-seq. Whole genome sequence (WGS) data of the *P. falciparum* W2 strain reported in [10] under SRA accession number SRR1335983 were analyzed using tools hosted by the public server at usegalaxy.org [103]. The WGS data were pre-processed using the Trimmomatic tool, aligned to the *P. falciparum* 3D7 v3.2 reference genome and variants called following the recommendations for haploid genomes [104]. Raw variants were filtered to remove indels, and single nucleotide polymorphisms (SNPs) with more than one alternative base, spanning more than one position e.g., AA–AT, and with variant quality score (QUAL) less than 40. 32,060 SNPs remained after filtering. The filtered SNPs were used with the *P. falciparum* 3D7 v3.2 reference genome to generate a genome index with HISAT2. Ribo-seq data were obtained from NCBI via accession numbers SRR1378560–SRR1378563 (run1) and SRR2014726–SRR2014729 (run2). Data from merozoite stage parasites were not analyzed as ribosome footprints were of much lower density overall for this stage [10]. Matching short-read RNA-Seq data from the same biological samples were obtained from NCBI via accession numbers SRR1378555–SRR1378558 (run1) and SRR2014721–SRR2014724 (run2). The raw data were processed using the fastp tool with the following options: adapter sequence specified (-a CTGTAGGCACCATCAATTCGTATGCCGTCTTCTGCTTG) and trimmed read length limited from 18 to 30 bp (-l 18 -b 30). Reads matching *P. falciparum* rRNA and tRNA were filtered out from processed reads using the sortmerna2.1b tool [105]. Filtered reads were aligned to the *P. falciparum* W2 genome index using HISAT2 guided by the *P. falciparum* 3D7 reference genome annotation with maximum intron 5 kb. Aligned reads from replicates of the same developmental stage were merged into one.bam file with SAMtools. To obtain read counts for the features of interest, simplified annotation files (SAF) were created for ORFs and downstream 3′-UTRs. We generated SAFs for annotated protein ORFs in reference transcripts (S2A Table) and altORFs in intergenic and antisense novel isoforms (S2B Table). 3′-UTR features accompanying ORFs and altORFs were defined as sequence from the stop codon to the 3′ end of the same RNA. SAFs and merged.bam files were used to obtain counts of mapped reads using the featureCounts tool with options selected for single, strand-specific reads (-s 1) and counting only reads with 100% feature overlap (—fracOverlap 1). Translational efficiency (TE) was determined from the ratio of normalized counts (ribo-seq: RNA-Seq). Counts were normalized by reads per kilobase per million mapped reads (rpkM) accounting for the length of the feature and the total number of mapped reads in the merged.bam file. Ribosome release score (RRS) was determined from the ribo-seq and RNA-Seq counts in ORF and downstream 3′-UTR features, as described previously [43]. TE and RRS were not determined for any ORF with zero counts, or zero counts in the accompanying 3′-UTR, or with a 3′-UTR shorter than 50 bp. TE and RRS values in at least one developmental stage were determined for 3,565 ORFs in reference transcripts, 246 altORFs in antisense isoforms, and 106 altORFs in intergenic isoforms. The TE and RRS values were $\log_2$ transformed and plotted using the ggplot2 R package. Kolmogorov-Smirnov two-sided, two-group tests comparing the distributions of TE and RRS were performed using the stats package in R. Holm-Bonferroni adjusted $P < .05$ was considered significant.

## Supporting information

**S1 Fig. SDS-PAGE analysis of HseIF4E-eIF4G_x6His recombinant 5′ cap binding protein.** Recombinant protein was expressed in *Escherichia coli* BL21(DE3) transformed with HseIF4E-eIF4G_x6His plasmid. Protein samples were separated by 12% SDS-PAGE and stained with Coomassie blue. (**A**) 5′ cap-binding assay by specific elution from aminophenyl-m⁷GTP (C10-spacer) agarose (Jena Bioscience). The migrations of prestained protein marker bands

(Thermo Scientific) are indicated to the left of the lane marked M. Lane 1, crude soluble protein extract; lanes 2–5, m⁷GTP eluted fractions; lane 6, void volume from NAP-25 Sephadex gel-filtration column; lanes 7–9, fractions 1–3 eluted from gel filtration column. The protein band of the size expected for HseIF4E-eIF4G_x6His fusion protein (46 kDa) is marked by an arrow. **(B)** HisTrap FF Ni²⁺-Sepharose purification of HseIF4E-eIF4G_x6His recombinant protein. Lane C, crude soluble extract; lane FT, flow-through of unbound protein; lane M, PageRuler Plus prestained protein marker (Thermo Scientific); lanes E1–E20, imidazole gradient elution fractions. HseIF4E-eIF4G_x6His protein (46 kDa, arrowed) eluted in fractions E2 and E3. **(C)** Q-Sepharose FF purification. Fractions E2 and E3 from HisTrap FF Ni²⁺-Sepharose were applied to the column. Lane M, PageRuler Plus prestained protein marker (Thermo Scientific); lanes 2–26, NaCl gradient elution fractions. HseIF4E-eIF4G_x6His protein (46 kDa, arrowed) eluting in fractions 12–15 was used for the enrichment of 5′ capped mRNA.
(TIF)

**S2 Fig. Isolation of *Plasmodium falciparum* nuclei.** Nuclei were isolated from GAL4-GFP *P. falciparum* transgenic parasites [80] by differential lysis and centrifugation. Four different fields of view are shown in separate rows, with columns labeled A–E showing different signals of the same field of view. **(A)** channel 1, Hoechst 33342 signal; **(B)** channel 2, GFP signal; **(C)** merged channel 1 and 2; **(D)** brightfield and **(E)** merged channel 1,2 and brightfield. Scale bars (2, 2, 2, and 1 μm) are shown in column **(E)**.
(TIF)

**S3 Fig. Construction of *Plasmodium falciparum* 3D7 isoform catalog (raw data to merged candidate isoforms).** Flowchart shows the data analysis steps used to construct the isoform catalog from raw long-read RNA-Seq data to merged candidate non-redundant isoforms. Raw data sources are shown as P (PacBio data from this study), PE (PacBio data from the Yang et al study [21]), N (Nanopore data from this study), and NE (direct RNA Nanopore data from the Lee et al study [22]). Figure was created using Diagrams.net (available from https://app.diagrams.net/).
(TIF)

**S4 Fig. Construction of *Plasmodium falciparum* 3D7 isoform catalog (filtering of merged candidate isoforms to final catalog).** Flowchart shows the final data analysis steps used to construct the isoform catalog. The TAMA merge file outputs of source information (TAMA_trans_report.txt) and merged candidate non-redundant isoforms (TAMA_merge.gtf) were used to construct the final isoform catalog. Figure was created using Diagrams.net (available from https://app.diagrams.net/).
(TIF)

**S5 Fig. Codon composition analysis.** Plots show the percent frequency of each codon in analysis windows (selected *Plasmodium falciparum* 3D7 v3 genomic positions, 10 bases upstream and downstream) in annotated protein-coding sequence regions. Selected positions corresponding to the 5′ end positions of novel isoforms assigned to the C2 cluster (C2 5′ end) are in blue and randomly selected positions from the same genes are shown in red.
(TIF)

**S6 Fig. Empirical cumulative distribution frequency plots of codons.** The plots of all 64 codons are shown for analysis windows corresponding to *Plasmodium falciparum* 3D7 C2 5′ ends (blue) and randomly selected positions (red). The distributions were compared by two-sample, two-sided Kolmogorov-Smirnov test; unadjusted test *P*-values are shown on each plot

and Holm-Bonferroni adjusted *P* < .05 are indicated in magenta.
(TIF)

**S7 Fig. Examples of genes with isoforms coding for truncated proteins.** Screenshots from the IGV genome browser [38] are shown for selected regions of *Plasmodium falciparum* 3D7 chromosome 10 (enolase gene top), and 9 (RhopH2 gene middle and RhopH3 gene bottom). Truncated protein isoforms expressed from these genes were reported previously from two-dimensional differential gel electrophoresis [65]. The structures of PlasmoDB [16] annotated genes and catalog isoforms are shown in the "Annotated genes" and "Catalog isoforms" tracks, respectively (boxes, exons; thick boxes, open reading frames; lines, introns, and arrows, strand orientation). In the "Catalog isoforms" track, reference transcripts are in blue and novel isoforms are in red. The color hue of each isoform reflects the level of support from long-read RNA-seq data (darker, more support). Isoform identifiers (S2 Table) are indicated after the gene identifier.
(TIF)

**S1 Table. Summary statistics of long-read RNA-Seq data.** Long read RNA-Seq data were obtained from four studies as indicated in row 1: Lee et al [22], Yang et al [21], this study, and Chappell et al [15]. The data were obtained from independent biological samples of *Plasmodium falciparum* indicated in row 2. Data from samples 1–10 were used to construct the transcriptomic catalog. Details of developmental stage and RNA fraction for each sample are shown in rows 3 and 4, respectively. Long-read sequencing platform is indicated in row 5. Processed read datasets obtained from each sample are indicated in row 6; datasets from PacBio sequencing have "lq" or "hq" suffixes to indicate low quality or high-quality assignment from Iso-Seq3 read processing. The dataset labels shown in this row are consistent with dataset source labels in S2 Table (columns 33, 36−38). The numbers of processed reads available for alignment in each dataset are shown in row 7. Row 8 shows the number of reads aligned to the *P. falciparum* 3D7 v3.2 reference genome [14] by minimap2 [84]. Row 9 shows the number of candidate non-redundant isoforms assigned by TAMA collapse [27] from the aligned reads for each dataset.
(XLSX)

**S2 Table. Transcriptomic catalog isoform classification.** The *Plasmodium falciparum* 3D7 transcriptomic catalog constructed from long-read sequencing data is shown for 5,568 reference transcripts (S2A) and 12,495 novel isoforms (S2B). Details are shown for the classification.txt output from SQANTI3 (columns 1−32, headers in orange; for explanation of column headers, see S1 & S2 Tables in Tardaguila et al [35]). The tama_merge_trans.txt output from TAMA merge [27] is shown in columns 33−38 (headers in blue). Column 33 (sources) refers to candidate non-redundant isoforms assigned by TAMA collapse [27] for each long-read RNA-Seq dataset for alignment described in S1 Table. Columns 34 (start_wobble_list) and 35 (end_wobble_list) refer to the "wobble", or range of exon start and end coordinates, respectively among candidate isoforms from different data sources merged into the final isoform. Columns 36 (exon_start_support) and 37 (exon_end_support) list the source data information of candidate isoforms used for assigning the 5′ and 3′ ends, respectively of the final merged isoform. Column 38 (all_source_trans) lists all candidate isoforms and data sources used for merging.
(XLSX)

**S3 Table. Isoform events in the transcriptomic catalog.** Isoform events identified by CAT-ANA [97] among reference transcripts (S3A) and novel isoforms (S3B). Event identifier is shown in column 1 (event). Isoforms inclusive (column 2) and exclusive (column 3) of the

event are indicated by catalog isoform identifiers (S1 Dataset and S2 Table). Column 4 (code) refers to the exon usage pattern among isoforms. Column 5 (strand) indicates which annotated genome strand (+ or–) the isoforms map to.
(XLSX)

**S4 Table. Significant 6-mers in the vicinity of reference transcript termination sites.** Tabulated results from kpLogo analysis [36] are shown for the most significant 6-mer at each position from 50 bp upstream (−50) to 45 bp downstream (45) of *Plasmodium falciparum* 3D7 reference transcript termination sites (TTS). 5,560 genomic sequences in the vicinity of TTS (test) and random (background) positions were used for analysis. Results shown from column 1 to 10: 6-mer sequence, nucleotide position relative to TTS, shift, enrichment test statistic, *P*-value (−log10 transformed), bonferroni-corrected *P*-value (−log10 transformed), the observed fraction of input sequences with this k-mer at this position allowing this amount of shift, the expected fraction of sequences with this k-mer at this position allowing this amount of shift, ratio (column 7 divided by column 8), and column 7 divided by the average of other positions.
(XLSX)

**S5 Table. Proteins encoded by novel isoforms.** Encoded proteins 29 residues or greater from all open reading frames among 12,495 novel isoforms were queried against protein databases. Proteins are grouped according to BLASTp results: no hit (S5A); *Plasmodium falciparum* protein hit (S5B); protein hit to other taxa (S5C). Query protein information is shown in columns 1–6 with gray headers. The identifier and SQANTI3 classification (S2B Table) of the isoform encoding the protein are shown in columns 1–3. The protein identifier, amino acid sequence, and number of residues are shown in columns 4–6, respectively. Top-ranked BLASTp hit details are shown for searches with the NCBI web application [40] against the non-redundant protein database (columns 7–12 with red headers) and the *P. falciparum* 3D7 protein database (columns 13–17 with blue headers). Search results with the eggNOG web application [41] are shown in columns 18–33 with purple headers.
(XLSX)

**S6 Table. Kolmogorov-Smirnov tests comparing distributions of translational efficiency and ribosome release score.** *P*-values and Holm-Bonferroni adjusted *P*-values are shown from two-sample two-sided Kolmogorov-Smirnov tests comparing distributions of $\log_2$ translational efficiency (TE) and $\log_2$ ribosome release score (RRS) among open reading frames in *Plasmodium falciparum* 3D7 reference transcripts, intergenic and antisense isoforms (Fig 8B). TE and RRS metrics were determined from ribo-seq data of ring, early trophozoite, late trophozoite and schizont sampled stages of development [10]. Pairwise comparisons were made among isoform groups for each metric and stage of development as indicated in columns 1–4. Unadjusted *P*-values shown as 2.20E-16 represent the minimum *P*-value that can be reported in R [95].
(XLSX)

**S7 Table. Polycistronic isoforms.** Novel isoforms completely spanning two or more annotated *Plasmodium falciparum* 3D7 genes were identified with BEDTools [92]. The isoforms were assigned to 114 non-overlapping genomic regions indicated in column 1. Isoform identifier (S1 Dataset and S2B Table) is shown in column 2. Information of annotated genes overlapping the isoform is shown in columns 3–15. Column 16 indicates whether polycistronic mRNA was reported previously from the same genomic region in Shaw et al [12].
(XLSX)

**S8 Table. Nearest annotated feature to intergenic isoforms.** Nearest annotated *Plasmodium falciparum* 3D7 features to novel isoforms classified as intergenic were identified with

BEDTools [92]. Genomic annotations of the isoform and the nearest genomic feature are shown in columns 1–5 and 6–11, respectively. The distance (nt) from the isoform to the nearest genomic feature (irrespective of strand) is shown in column 12. Description of the nearest feature is shown in columns 13–15.
(XLSX)

**S9 Table. Oligonucleotide sequences.**
(XLSX)

**S1 Dataset. Transcript annotation file.** Reference and novel isoform transcript annotations in.bed12 format associated with the *Plasmodium falciparum* 3D7 v3.2 reference genome [14] (Fields 1, 2, 3, 6, 7, 8, 10, 11 & 12). Field 4 contains the isoform identifier and PlasmoDB [16] associated gene as shown in S2 Table. Field 5 contains an arbitrary score denoting the degree of support from long-read RNA-Seq datasets from 166 (2 datasets) to 945 (18 datasets). Field 9 contains a color code denoting reference transcript (0, 0, 255) or novel isoform (255, 0, 0).
(BED)

**S1 Text. HseIF4E-eIF4G_x6His plasmid sequence.**
(DOCX)

# Acknowledgments

We thank Jim Blackburn for RNA sequins, Khi Pin Chua (PacBio Singapore Pte. Ltd.) and Mengquan Yang for assistance with PacBio data analysis, and Manolis Maragkakis for assistance with codon composition analysis.

# Author Contributions

**Conceptualization:** Philip J. Shaw, Sissades Tongsima, Jittima Piriyapongsa.

**Data curation:** Philip J. Shaw, Pavita Kaewprommal, Chayaphat Wongsombat, Chumpol Ngampiw, Tana Taechalertpaisarn, Jittima Piriyapongsa.

**Formal analysis:** Philip J. Shaw, Pavita Kaewprommal, Jittima Piriyapongsa.

**Funding acquisition:** Philip J. Shaw, Sumalee Kamchonwongpaisan, Jittima Piriyapongsa.

**Investigation:** Philip J. Shaw, Chumpol Ngampiw, Tana Taechalertpaisarn, Jittima Piriyapongsa.

**Methodology:** Philip J. Shaw, Pavita Kaewprommal, Chayaphat Wongsombat, Tana Taechalertpaisarn, Jittima Piriyapongsa.

**Project administration:** Philip J. Shaw, Sumalee Kamchonwongpaisan, Sissades Tongsima, Jittima Piriyapongsa.

**Resources:** Sissades Tongsima.

**Software:** Pavita Kaewprommal, Chumpol Ngampiw.

**Supervision:** Philip J. Shaw, Sumalee Kamchonwongpaisan, Sissades Tongsima, Jittima Piriyapongsa.

**Visualization:** Philip J. Shaw.

**Writing – original draft:** Philip J. Shaw, Jittima Piriyapongsa.

**Writing – review & editing:** Philip J. Shaw, Sumalee Kamchonwongpaisan, Sissades Tongsima, Jittima Piriyapongsa.

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
