## [Decision Letter · Decision Letter 0]

6 Jul 2022

PONE-D-22-13960Transcriptomic complexity of the human malaria parasite Plasmodium falciparum revealed by long-read sequencingPLOS ONE

Dear Dr. %Shaw%,

Thank you for submitting your manuscript to PLOS ONE. After careful consideration, we feel that it has merit but does not fully meet PLOS ONE’s publication criteria as it currently stands. Therefore, we invite you to submit a revised version of the manuscript that addresses the points raised during the review process.

In the reviewer's comments, you will notice that while the reviewers have found your study novel and adding to our current knowledge on transcriptomic plasticity of P. falciparum parasites, they have raised a few major concerns and put forward some useful suggestions (such as wet-lab validations of transcript isoforms) to improve the quality of this manuscript. Please address these concerns while revising the manuscript.

We look forward to receiving your revised manuscript.

Kind regards,

Arnab Pain, Ph.D.

Academic Editor

PLOS ONE

Journal Requirements:

Reviewers' comments:

Reviewer's Responses to Questions

**Comments to the Author**

1. Is the manuscript technically sound, and do the data support the conclusions?

Reviewer #1: Partly

Reviewer #2: Yes

Reviewer #3: Partly

Reviewer #4: Yes

2. Has the statistical analysis been performed appropriately and rigorously? 

Reviewer #1: N/A

Reviewer #2: N/A

Reviewer #3: No

Reviewer #4: Yes

3. Have the authors made all data underlying the findings in their manuscript fully available?

Reviewer #1: No

Reviewer #2: Yes

Reviewer #3: No

Reviewer #4: Yes

4. Is the manuscript presented in an intelligible fashion and written in standard English?

Reviewer #1: Yes

Reviewer #2: Yes

Reviewer #3: Yes

Reviewer #4: Yes

5. Review Comments to the Author

Reviewer #1: It is generally known that genes are transcribed in different pattern resulting isoforms. Plasmodium falciparum is one of the most important pathogens and model strain of Apicomplexan parasites as well. However, its isoforms in transcripts has not described enough. Here, the authors provided a comprehensive catalog of the isoforms using long-read sequencing.

To obtain full-length complete cDNA, they utilized an improved eIF4E system for capturing of 5’ capped mRNA. Analytical pipeline for identification of isoforms were optimized using sequin RNA. Comprehensive TSS data sets in public were utilized for filtering possible artificially truncated isoforms. Those started at low chromatin density region were excluded as well. Resulting isoforms were compared with the standard gene model then 12,495 novel isoforms were identified.

The provided gene model including multiple isoforms are quite informative and will contribute for further analyses in the parasites biology after substantial revision.

Major points:

1.

Line 28, alternative 5’ and 3’ are not a part of alternative splicing. It is better to use “isoform” instead of alternative splicing. Line 59 and others as well.

2.

The gtf you provide will be quite useful information for further studies in research community.

However, there are some lack in information for convenient use. If you provide additional gff/gtf integrating the standard gff in PlasmoDB into yours, it will be much beneficial. In the gff/gtf, differentiate your sequences from the PlasmoDB gff and previous studies (maybe using column 2 “source”), add gene description to yours as well, and select and note representative one if gene has multiple isoforms. It might be decided by number of sequence reads.

3.

Line 287 and Fig 1A, almost 50% reads were truncated at their 5’ terminal. How did it happen? Was it caused by insufficient optimization in analytical pipeline or TSS enrichment using the eIF4 system? According to the reference #17 which applied the similar system, distribution of TSSs are around 100 bp. There are other TSS specific methods such as CAGE and oligo-capping. Have you compared the performance among them and described why you used the eIF4 system among them?

4.

Making transcriptomic catalog is one of the major achievements of this study; however, the process is complicated to understand at a glance. Please provide a flowchart as a supplemental figure.

5.

Line 316 and Fig 3A, have you include or exclude the C2 isoforms from the final transcriptomic catalog? Besides, if C2 5’ ends are less likely true TSS, how were these caused? You have described it in the following paragraph (line 319-329); however, co-translational chevage generates cap less RNA then the eIF4 system excludes the RNA; therefore, 5’ end generated by co-translational chevage should not be observed in your study.

6.

Line 190, what is the 37 different 6-mers? Also, it seems that there are a position constrained conserved sequence from -44 to -26. Have you searched any motifs using other tools such as MEME?

7.

Deposit the sequence data to INSDC (International Nucleotide Sequence Database Collaboration).

Minor points:

1.

Line 205-208, I can not find any difference between spliced and retained.

2.

Fig 1C, Fig 3A, Fig 5A and B, and Fig 8, what does “density” in the vertical axis means?

3.

S1 Table, need more descriptions. For example, what is the directRNA, Yang1hq, ringtotalhq, NPcDNA, and Chappell PacBio RS? The other supplemental tables lack sufficient information as well.

Reviewer #2: In the manuscript entitled “Transcriptomic complexity of the human malaria parasite Plasmodium falciparum revealed by long-read sequencing”, authors attempted to improve the P. falciparum transcriptomic catalog by producing the novel gene isoforms and rigorously analyzed the different Alternative splicing events observed in the cDNA sequences. The manuscript is written and organized in a systematic manner, especially materials and methods. However, I have few minor comments and request authors to address these in the next version of the manuscript.

o Line 140: "9,004 of the 12,495 novel isoforms were supported by data…” Please specify the why authors are calling these transcripts as novel even though they are reported in previous studies. Definition of term ‘novel’ is confusing here.

o Authors must recommend how the novel datasets (isoforms) presented in this study can further be used for the subsequent analysis, especially from the bioinformatics perspective.

o Please add a study limitation sub-section in the discussion section.

o Are these isoforms and AS sites is specific to IDC stage only. Please discuss.

Reviewer #3: The manuscript addresses one of the major challenges in transcript annotation for P falciparum but needs to address the following issues specifically an experimental validation of the results using isoform specific qCPR and the details of the analyses performed outlined below.

Major points:

Every statement claiming significant differences should accompany the appropriate statistical test.

1. The enrichment of the new cap binding protein should be demonstrated using a qPCR and the statement five-fold better enrichment should be statistically tested.

2. 2. Line 129: How was the TAMA tool selected over the other tools should be discussed in detail by statistically substantiating the statement "these tools falsely assigned isoforms from misaligned reads to non-exonic regions including antisense". For example - What percentatge of the reads were falsely assigned? Was the false assignment significantly different statistically?

3. Line 135: Please detail how were the isoforms combined ? Were they collapsed if they were within xxx bases at either ends ?

Line 138: "lacking supporting evidence of 5′ capped end" - What dataset was used as supporting evidence for 5' capped ends. The strategy used by Adjalley et al 2016 is enzymatic while the one used by authors of the current group is pull-down based. How are the two kinds of datasets reconcilled?

4. Line 207-208: "higher percent GC (Fig 5A) and shorter length (Fig 5B) compared with spliced introns; however, it

should be noted that the mean differences are small." Please add numbers and appropriate tests to support these statements.

5. Does the analysis of ATSS and ATTS clusters change if only the consensus isoforms (isoforms with support from other studies) are used

6. An isoform specific qPCR is a must to validate the results which could be done on retained introns isoforms or stage specific isooforms.

7. The authors state that both PacBio and Nanopore sequencing was used, but a comparison between of data generated using the two techniques is currently missing. Given that the data is not always concordant such an analysis is paramount to generate a viable annotation.

Minor points

1. In multiple places the statement "we utilized long-read platforms to sequence cDNA of P. falciparum IDC stages" is used. This should mention what platforms and which stages clearly. Also, the number of samples (replicates) read depths, coverage and other sequencing QC information should be clearly mentioned in the text.

2. Line 141: "9,004 of the 12,495 novel isoforms were supported by data from other studies in combination with data from this study", supported by atleast one oher study or all the studies

3. Line 155-158: Please indicate the percentages of different events in the text as well.

4. Line 166: Please give reference of the H2A.Z dataset

5. This study will be extremely useful to the community if an updated gtf file is provided with new isoforms along with the level of support for it ( ex. supported by 2 studies, 3 studies etc)

6. Manuscript should have a data availability statement.

Reviewer #4: Shaw et al., in the current manuscript have catalogued additional alternative splicing (AS) events in P. falciparum IDC stages using the long read next generation sequencing technologies like PacBio and Nanopore platforms. Of the 12,495 novel mRNA isoforms detected by them, majority were either alternative 5' or 3' events. They have also identified a good number of intron retention events. This study further provides information about the in-silico identification of additional proteins that adds to the current proteome repertoire. Long read sequencing technologies identify AS events more accurately and hence this study provides important information related to unidentified AS events that can assist in understanding gene and protein regulation in the deadly malaria parasite P. falciparum.

Minor comments:

1. In case of intron retention events that were reported, it would have been great to validate a couple of intron retention events using reverse transcriptase quantitative PCR by targeting primers to the retained intron region.

2. I understand that alternative ORFs can code for variant proteins that have not been annotated yet. I will be happy to see if the authors can validate a few intergenic isoforms for their ability to code proteins. This may also include a couple of isoforms that span two or more genes.

6. PLOS authors have the option to publish the peer review history of their article (what does this mean?). If published, this will include your full peer review and any attached files.

Reviewer #1: No

Reviewer #2: No

Reviewer #3: No

Reviewer #4: **Yes: **AMIT KUMAR SUBUDHI

---

## [Author Response · Author response to Decision Letter 0]

5 Sep 2022

PONE-D-22-13960

Transcriptomic complexity of the human malaria parasite Plasmodium falciparum revealed by long-read sequencing

PLOS ONE

Dear Dr. Shaw,

Thank you for submitting your manuscript to PLOS ONE. After careful consideration, we feel that it has merit but does not fully meet PLOS ONE’s publication criteria as it currently stands. Therefore, we invite you to submit a revised version of the manuscript

that addresses the points raised during the review process.

In the reviewer's comments, you will notice that while the reviewers have found your study novel and adding to our current knowledge on transcriptomic plasticity of P. falciparum parasites, they have raised a few major concerns and put forward some useful suggestions (such as wet-lab validations of transcript isoforms) to improve the quality of this manuscript. Please address these concerns while revising the manuscript.

If applicable, we recommend that you deposit your laboratory protocols in protocols.io to enhance the reproducibility of your results. Protocols.io assigns your protocol its own identifier (DOI) so that it can be cited independently in the future. For instructions see: https://journals.plos.org/plosone/s/submission-guidelines#loc-laboratory-protocols. Additionally, PLOS ONE offers an option for publishing peer-reviewed Lab Protocol articles, which describe protocols hosted on protocols.io Read more information on sharing protocols at https://plos.org/protocols?utm_medium=editorial-email&utm_source=authorletters&utm_campaign=protocols.

We look forward to receiving your revised manuscript.

Kind regards,

Arnab Pain, Ph.D.

Academic Editor

PLOS ONE

Journal Requirements:

Reviewers' comments:

Reviewer's Responses to Questions

Comments to the Author

1. Is the manuscript technically sound, and do the data support the conclusions?

Reviewer #1: Partly

Reviewer #2: Yes

Reviewer #3: Partly

Reviewer #4: Yes

2. Has the statistical analysis been performed appropriately and rigorously? 

Reviewer #1: N/A

Reviewer #2: N/A

Reviewer #3: No

Reviewer #4: Yes

3. Have the authors made all data underlying the findings in their manuscript fully available?

Reviewer #1: No

Reviewer #2: Yes

Reviewer #3: No

Reviewer #4: Yes

4. Is the manuscript presented in an intelligible fashion and written in standard English?

Reviewer #1: Yes

Reviewer #2: Yes

Reviewer #3: Yes

Reviewer #4: Yes

5. Review Comments to the Author

Reviewer #1: It is generally known that genes are transcribed in different pattern resulting isoforms. Plasmodium falciparum is one of the most important pathogens and model strain of Apicomplexan parasites as well. However, its isoforms in transcripts has not described enough. Here, the authors provided a comprehensive catalog of the isoforms using long-read sequencing.

To obtain full-length complete cDNA, they utilized an improved eIF4E system for capturing of 5’ capped mRNA. Analytical pipeline for identification of isoforms were optimized using sequin RNA. Comprehensive TSS data sets in public were utilized for filtering possible artificially truncated isoforms. Those started at low chromatin density region were excluded as well. Resulting isoforms were compared with the standard gene model then 12,495 novel isoforms were identified.

The provided gene model including multiple isoforms are quite informative and will contribute for further analyses in the parasites biology after substantial revision.

Response: We thank the reviewer for the succinct summary. However, we wish to correct potential misunderstanding in the sentence: “Those started at low chromatin density region were excluded as well.” We did not filter isoforms using chromatin density data.

Major points:

1.

Line 28, alternative 5’ and 3’ are not a part of alternative splicing. It is better to use “isoform” instead of alternative splicing. Line 59 and others as well.

Response: We agree that alternative splicing events are distinct from alternative start and termination sites. To avoid potential confusion, we changed “alternative splicing” to “isoform” or “isoform events” as suggested in the revised manuscript. 

2.

The gtf you provide will be quite useful information for further studies in research community.

However, there are some lack in information for convenient use. If you provide additional gff/gtf integrating the standard gff in PlasmoDB into yours, it will be much beneficial. In the gff/gtf, differentiate your sequences from the PlasmoDB gff and previous studies (maybe using column 2 “source”), add gene description to yours as well, and select and note representative one if gene has multiple isoforms. It might be decided by number of sequence reads.

Response: 

We agree with the reviewer that a single comprehensive genome annotation file incorporating the new transcriptomic information consistent with the PlasmoDB format would be of most benefit to the community. However, the task of revising the genome annotation is best left to the PlasmoDB curators. Rather than attempting to make a new, comprehensive genome annotation file, we decided to make a simpler transcript annotation file with features that researchers may find useful for transcriptomic studies to complement the current PlasmoDB genome annotation. 

In the revised manuscript, we replaced the S1_dataset.gtf file with a more compact S1_dataset.bed file. The revised annotation file in .bed12 format includes reference transcripts and novel isoforms assigned in S2 Table. The fourth field of the S1_dataset.bed file includes the isoform identifier shown in S2 Table and gene identifiers in the current PlasmoDB nomenclature, e.g. PF3D7_0417200. Therefore, gene descriptions and other information of interest can be obtained by cross-referencing with PlasmoDB. Reference transcripts have a blue color code, whereas novel isoforms are coded in red. Every isoform also has a score to denote the level of support from long-read RNA-Seq datasets as shown in S2 Table (column 33). When displayed on genome browser software such as IGV, the scores indicating the level of support are displayed as different color hues, with darker hues indicating greater levels of support. Note that all mRNA reference transcripts in the current PlasmoDB genome annotation are present in the revised S1_dataset.bed file, including reference transcripts for which there are no supporting long-read RNA-Seq data. 

3.

Line 287 and Fig 1A, almost 50% reads were truncated at their 5’ terminal. How did it happen? Was it caused by insufficient optimization in analytical pipeline or TSS enrichment using the eIF4 system? According to the reference #17 which applied the similar system, distribution of TSSs are around 100 bp. There are other TSS specific methods such as CAGE and oligo-capping. Have you compared the performance among them and described why you used the eIF4 system among them?

Response: A major difficulty for transcript isoform annotation is the noise of reads with truncated 5′ ends, which is addressed in detail in the Kuo et al papers (originally cited references 26 and 41). Truncated cDNA (cDNA that does not extend to the original mRNA 5′ end) occurs because of pausing at RNA secondary structure and degradation of RNA template during the reverse transcription reaction (Das et al 2001, DOI: 10.1152/physiolgenomics.2001.6.2.57). Although we enriched 5′ capped mRNA using the HseIF4E-eIF4G_x6His fusion protein, we cannot prevent the truncated cDNA products of reverse-transcription. We tried enrichment of full-length cDNA with the HseIF4E-eIF4G_x6His fusion protein after the cDNA synthesis step (Shaw et al., 2021; originally cited reference 17), but the yield was too low to be practical for constructing long-read RNA-Seq libraries. The CAGE method mentioned by the reviewer is currently only employed for short-read RNA-Seq and captures only 5′ terminal fragments of full-length cDNA via the Cap-Trapper enrichment strategy. To our knowledge, there is no published long-read RNA-Seq protocol based on the CAGE method for enriching full-length cDNA representing complete mRNAs (from the original 5′ to 3′ ends), although an unpublished preprint describes a long-read RNA-Seq protocol with Cap-Trapper enrichment in development (Pardo-Palacios et al 2021; DOI: 10.21203/rs.3.rs-777702/v1). The oligo-capping method mentioned by the reviewer is also inefficient, although it has been used in a few studies in other organisms, e.g., chicken (Kuo et al 2020, originally cited reference 41). The currently commercially available Teloprime kit (Lexogen company) for enriching full-length cDNA after the cDNA synthesis step is biased towards short cDNA (Cartolano et al., 2016; originally cited reference 78), and our preliminary testing of the Teloprime kit with P. falciparum samples showed strongly biased representation of short cDNA <1 kb, which was communicated to the manufacturer. 

Instead of using an inefficient enrichment method for full-length cDNA after the cDNA synthesis step, we chose to enrich mRNA with a 5′ cap binding protein to increase representation of isoforms with alternative 5′ capped ends. We filtered isoforms bioinformatically using the 5′ cap signals from short-read RNA-Seq to eliminate truncated cDNA artifacts. This approach has been used in other long read RNA-Seq studies, e.g., the rat hippocampus (Wang et al 2019; DOI: 1038/s41467-019-13037-0), and is an integral part of the SQANTI isoform filtering pipeline (Tardaguila et al., 2018; originally cited reference 34). We appreciate the reviewer’s concerns over our experimental protocol for enriching 5′ capped mRNA. However, we think that extensive comparison of methods for enrichment of full-length cDNA would detract from the story and so prefer not to add this information to the manuscript. 

4.

Making transcriptomic catalog is one of the major achievements of this study; however, the process is complicated to understand at a glance. Please provide a flowchart as a supplemental figure.

Response: we added flowchart diagrams describing the catalog construction process as suggested by the reviewer in new Supplementary figures S3 and S4 in the revised manuscript.

5.

Line 316 and Fig 3A, have you include or exclude the C2 isoforms from the final transcriptomic catalog? Besides, if C2 5’ ends are less likely true TSS, how were these caused? You have described it in the following paragraph (line 319-329); however, co-translational chevage generates cap less RNA then the eIF4 system excludes the RNA; therefore, 5’ end generated by co-translational chevage should not be observed in your study.

Response: we included all isoforms after filtering in the final catalog, including those with C2 5′ ends. We did not address how C2 5′ ends were generated in this manuscript, but speculated that they are generated post-transcriptionally. According to the available information in other eukaryotes, we conjecture that 5′ capping machinery is also present in the parasite cytoplasm which can add a new 5′ cap structure to co-translationally cleaved mRNA in a process described as mRNA re-capping. This process is reviewed by Trotman and Schoenberg (originally cited reference 31), which was first mentioned in the results section of the original manuscript (line 176−179). In the revised manuscript, we clarified the Discussion section by adding that isoforms with C2 5′ ends could be generated post-transcriptionally by re-capping after co-translational cleavage (line 351−353; please refer to the line numbering in the clean Manuscript file). 

6.

Line 190, what is the 37 different 6-mers? Also, it seems that there are a position constrained conserved sequence from -44 to -26. Have you searched any motifs using other tools such as MEME?

Response: 

We apologize for our oversight that all overrepresented 6-mers at each position upstream of the TTS are not obvious in Fig. 4B, in particular the 6-mers with low –log10(P) value that are displayed as small features on the figure. In the revised manuscript, we added a supplemental data file (new S4 Table) of all motifs reported by kpLogo, which are displayed graphically in Fig. 4B. We used the kpLogo tool as this is specifically designed for finding short motifs (2–6 nt). Our search was based on the assumption from other eukaryotes that polyadenylation signals constitute short motifs �6 nt (Neve et al 2017, originally cited reference 48). The MEME suite of tools is better suited for finding longer sequence motifs 6−50 nt in length (Bailey et al 2006; DOI: 10.1093/nar/gkl198). 

To answer the reviewer’s question, we tested the STREME and CentriMo programs in the MEME suite (available at https://meme-suite.org/meme/; last accessed 03/08/22) using the same sequences as used for kpLogo. These tools showed three motifs with significant (P<.05) positional enrichment in the vicinity of reference TTS, i.e., ATATATATAT, AAAAAAAAAAAAAAK, and AAAAAAAAA as shown below: 

Figure response to comment 6 from reviewer #1. 

CentriMo 5.4.1 display of positional enrichment for motifs identified as over-represented in 5560 P. falciparum 3D7 sequences in the vicinity of reference transcript termination sites (-50 to +50) compared with 5560 random genomic sequences of the same length. Over-represented motifs were first identified with STREME using settings: limit motifs to 10, align center, minimum word length 6, and maximum word length 15. The top five motifs found by STREME were: AAAAAAAAAAAAAAK (2551 occurrences), AAAAAAAAA (2116 occurrences), ATATATATAT (3840 occurrences), CTATTCTTYTAATAA (110 occurrences), and TAAAATAAAATA (278 occurrences). Positional enrichment of these motifs was assessed with CentriMo run in the enrichment mode with control random sequences as background, and searching given strand only with default settings. 

The AAAAAAAAAAAAAAK and AAAAAAAAA motifs are enriched in close proximity downstream of the TTS, whereas the ATATATATAT motif is enriched further downstream. As (AT) dinucleotides are generally present at higher frequencies in the more A/T rich intergenic regions of the P. falciparum genome, we think that the ATATATATAT motif found by the MEME suite is unrelated to transcriptional termination. The AAAAAAAAAAAAAAK and AAAAAAAAA motifs are supportive of the adenine-rich significant 6-mers found by kpLogo and the base composition plots shown in Fig. 4. In summary, the analysis by the MEME suite of tools supports our original conclusion that in P. falciparum has prominent adenine-rich regions in the vicinity of the TTS, but otherwise lacks distinct polyadenylation signals. 

7.

Deposit the sequence data to INSDC (International Nucleotide Sequence Database Collaboration).

Response: all raw sequence data are already deposited in the NCBI GEO under accession number GSE186109, as indicated in the “Data Availability statement” in the manuscript. The NCBI GEO database is a widely accepted resource for archiving of raw data and associated meta-data which are indexed, cross-linked and searchable (Barrett et al 2013; DOI: 10.1093/nar/gks1193). The raw sequencing data (SRA files) associated with the GEO accession will be made publicly available upon acceptance of the manuscript. 

Minor points:

1.

Line 205-208, I can not find any difference between spliced and retained.

Response: the distributions of intron length and %GC are significantly different by two-sample Kolmogorov-Smirnov two-sided test as indicated in the Fig. 5 legend. For clarity, we added that distributions were compared by Kolmogorov-Smirnov test in the body of the text in the revised manuscript (line 209). In the Discussion, we did emphasize that the differences in distributions are small and practically not sufficient to explain the biological phenomenon of intron retention. 

2.

Fig 1C, Fig 3A, Fig 5A and B, and Fig 8, what does “density” in the vertical axis means?

Response: 

The axes labelled “density” refer to the probability density function of the variable from the kernel density calculated by the ggplot2 R package. We added this information to the figure legends (revised Fig 1C, Fig 3A, Fig 5A and B, and Fig 8A). 

3.

S1 Table, need more descriptions. For example, what is the directRNA, Yang1hq, ringtotalhq, NPcDNA, and Chappell PacBio RS? The other supplemental tables lack sufficient information as well.

Response: We added descriptions for all rows and columns to the S1 Table and other Supplementary tables (S2−S8 Table). 

Reviewer #2: In the manuscript entitled “Transcriptomic complexity of the human malaria parasite Plasmodium falciparum revealed by long-read sequencing”, authors attempted to improve the P. falciparum transcriptomic catalog by producing the novel gene isoforms and rigorously analyzed the different Alternative splicing events observed in the cDNA sequences. The manuscript is written and organized in a systematic manner, especially materials and methods. However, I have few minor comments and request authors to address these in the next version of the manuscript.

o Line 140: "9,004 of the 12,495 novel isoforms were supported by data…” Please specify the why authors are calling these transcripts as novel even though they are reported in previous studies. Definition of term ‘novel’ is confusing here.

Response: Although our isoform catalog was created using previously published long-read RNA-Seq data in addition to our new data, surprisingly none of these earlier studies actually provided individual isoform structures in detail, e.g. as .gff/.gtf annotation data files and only representative examples of isoforms were illustrated in figures. Moreover, the P. falciparum transcriptome annotation in PlasmoDB (release 58: 23rd June 2022) has not been updated using these data. We included a definition of “novel isoforms” in the original Methods section (line 661−669), although we agree with the reviewer that this may be difficult for some to follow because of the obfuscating bioinformatic details. We revised the Results section with a simpler definition of novel isoforms as those “not currently annotated as transcripts in PlasmoDB” in the revised manuscript (line 135–136). 

o Authors must recommend how the novel datasets (isoforms) presented in this study can further be used for the subsequent analysis, especially from the bioinformatics perspective.

Response: The S1 Dataset, S2 Table and S3 Table are important resources for further bioinformatic study of isoforms. The translated protein sequences of the novel isoforms in S4 Table (S5 Table in revised manuscript) can be used in proteomic studies. We added this information to the concluding paragraph of the Discussion (line 455–462). 

o Please add a study limitation sub-section in the discussion section.

Response: in the revised manuscript, we included a “limitations of the study” sub-section as suggested (line 441−453). This section highlights the following limitations: lack of data for other stages of the life cycle, no information of isoform abundance, and protein-coding potential assessed only in intergenic and antisense isoforms. 

o Are these isoforms and AS sites is specific to IDC stage only. Please discuss.

Response: we identified structural categories of isoforms exclusively represented among PacBio datasets from different intra-erythrocytic (IDC) stages (Fig. 6), although we do not claim that isoforms and associated alternative transcript events are stage-specific, since this would require quantitative analysis of isoform abundance across the IDC stages. Quantitative analysis of isoform abundance from PacBio data is inherently challenging because the depth of sequencing is much lower than short-read RNA-seq, and the coverage of isoforms is non-uniform since the 5′ end is less likely to be covered by sequencing reads compared with the 3′ end. We are aware of a recent publication which addresses these challenges using an Expectation-Maximization approach for estimating the relative abundance of isoforms from PacBio data (Hu et al 2021; DOI: 10.1186/s13059-021-02399-8); however, this approach can only detect differential alternative splicing among two conditions and cannot quantify changes across multiple conditions, e.g., developmental time-points. Readers may find the TAMA merge output information in the S2 Table (column headers in blue) useful to identify individual isoforms that could be more abundant in one stage than another based on their representation in different RNA-Seq libraries. However, independent data of isoform abundance are needed to prove stage-specificity. We added this limitation to the “limitations of the study” discussion sub-section of the revised manuscript (line 441–453). If the reviewer is also asking whether the isoforms we reported are specific to the IDC stages, we cannot answer this as we don’t have long-read RNA-Seq data for comparison from other stages of the parasite life cycle, i.e., gametocyte, mosquito and liver stages. We highlighted the lack of data for these stages of the life cycle in the “limitations of the study” sub-section of the revised Discussion (line 441–453). 

Reviewer #3: The manuscript addresses one of the major challenges in transcript annotation for P falciparum but needs to address the following issues specifically an experimental validation of the results using isoform specific qCPR and the details of the analyses performed outlined below.

Major points:

Every statement claiming significant differences should accompany the appropriate statistical test.

Response: We apologize that for every statement of “significant differences” in the body of the text referring to a figure we did not include details of the statistical tests performed and P-values, which were indicated in the figure or figure legends. The details of all tests used, including thresholds for significance, are given in the materials and methods section. For clarity, we revised the manuscript by adding details of which test was performed to all statements of significance in the body of the text. For brevity and readability, we prefer to present P-values from test statistics in the figure (Fig 6; S6 Fig), figure legends (Fig 5; Fig 8A) or Supplementary Table (new S4 and new S6 Table). 

1. The enrichment of the new cap binding protein should be demonstrated using a qPCR and the statement five-fold better enrichment should be statistically tested.

Response: We provided evidence from 5′ cap binding assay (pull-down with aminophenyl-m7GTP (C10-spacer) agarose) in S1 Fig that the HseIF4E-eIF4G_x6His fusion protein has the expected binding activity for the 5′ cap structure on mRNA and we successfully used this protein to enrich P. falciparum mRNA for constructing sequencing libraries. Hence, we are satisfied that the HseIF4E-eIF4G_x6His fusion protein was a suitable tool for enriching 5′ capped mRNA to construct long-read RNA-Seq libraries. In the original manuscript, we paraphrased the claims from US patent 6,703,239 (Guegler et al 2004; originally cited reference 23) that eIF4E-eIF4G fusion protein has five-fold or better enrichment than eIF4E alone (line 107−108), but we did not intend to make a claim of enrichment efficiency specifically for the HseIF4E-eIF4G_x6His protein that we produced. 

As suggested by the reviewer, we tested the performance of the HseIF4E-eIF4G_x6His fusion protein for enrichment of 5′ capped transcripts by reverse-transcription quantitative PCR (RT-qPCR) assay. We synthesized cDNA from a sample of P. falciparum mixed-stage total RNA and enriched for cDNA derived from 5′ capped transcripts using recombinant 5′ cap binding protein immobilized on beads. We measured the abundance (Cq) of the cDNA made from lactate dehydrogenase (ldh) transcript normalized to that of an uncapped mitochondrial RNA (cox3) using primers and assay conditions described in Shaw et al 2007 (originally cited reference 22). The enrichment factor was determined as the normalized abundance (2�Cq) in enriched sample compared with the flow-through sample. The result (see below) shows that the HseIF4E-eIF4G_x6His protein has a greater enrichment factor than the GST-PfeIF4E originally described in Shaw et al 2007 (originally cited reference 22), although we did not replicate the experiment to determine whether the difference was significant. 

Figure response to point 1 reviewer #3. 

Enrichment of cDNA from a 5′ capped transcript (ldh) normalized to that of an uncapped transcript (cox3) was determined by reverse-transcription quantitative PCR. The cDNA from 5′ capped transcripts was enriched using 5′ cap binding proteins HseIF4E-eIF4G_x6His described in this manuscript and GST-PfeIF4E described previously (Shaw et al 2007). 

We do not want to make claims on whether the HseIF4E-eIF4G_x6His fusion protein is superior to other 5′ cap binding proteins for enrichment of 5′ capped mRNA, and so we changed the text to mention only that the fusion protein was used for enrichment of 5′ capped mRNA with reference to our previous work (revised manuscript line 100−103).

2. 2. Line 129: How was the TAMA tool selected over the other tools should be discussed in detail by statistically substantiating the statement "these tools falsely assigned isoforms from misaligned reads to non-exonic regions including antisense". For example - What percentatge of the reads were falsely assigned? Was the false assignment significantly different statistically?

Response: We used the same .SAM alignment file with all isoform annotation tools; therefore, the aligned reads are the same for all tools. The tools differed in how they assigned isoforms from the aligned reads. The FLAIR and stringtie2 tools assigned isoforms to misaligned reads that were not considered by TAMA collapse. This is because by default, the TAMA collapse tool only considers aligned reads with 1% or less of soft-clipped bases (bases at the end the read that are not part of the alignment, usually representing untrimmed adapter) and less than 15% mismatched bases in the part of the read aligned to the genome. In contrast, the stringtie2 and FLAIR tools use all aligned reads, including low-quality reads with many soft-clipped bases and mismatches. It is difficult to make a fair statistical test of tool performance, e.g., using evaluation metrics from Receiver Operator Characteristic analysis because the tools assign isoforms by different criteria. The FLAIR tool requires a reference transcript annotation and all isoforms not matching the reference transcripts are put into a separate category as “inconsistent isoforms”. In contrast, the TAMA collapse and stringtie2 tools assign isoforms in a manner naïve to reference annotation. From analysis of our data, TAMA collapse low and stringtie2 gave very similar performance overall with the latter assigning slightly more isoforms matching the ground truth, in agreement with the TAMA collapse publication (Kuo et al 2020; originally cited reference 26). However, we attached more importance to the isoforms identified by GffCompare in the low priority classes, i.e., greatest divergence from the ground truth as these isoforms are assigned from misaligned reads. From the analysis of RNA sequin data, we selected TAMA collapse low for assigning P. falciparum isoforms as this tool is least likely to assign false isoforms from misaligned reads, while still capable of detecting most of the expressed isoforms. In the revised manuscript, we added a sentence to clarify the choice of TAMA collapse low tool for assigning P. falciparum isoforms (revised manuscript 125–128). 

3. Line 135: Please detail how were the isoforms combined ? Were they collapsed if they were within xxx bases at either ends ?

Response: as stated in the materials and methods section (line 653), the settings for TAMA merge were: -a 300 -z 300 -m 20 -d merge_dup. Under these settings, isoforms with 5′ or 3′ ends within 300 bp of the common end for the group are collapsed. These settings were used in the Kuo et al 2020 paper (originally cited reference 26) describing the TAMA merge tool. To aid readers unfamiliar with the tool, we added a brief explanation of the TAMA merge settings used in the methods section of the revised manuscript (line 684−685). 

Line 138: "lacking supporting evidence of 5′ capped end" - What dataset was used as supporting evidence for 5' capped ends. The strategy used by Adjalley et al 2016 is enzymatic while the one used by authors of the current group is pull-down based. How are the two kinds of datasets reconcilled?

Response: 

We apologize for the unclear wording in the Methods section (line 639–647). We combined the 5′ cap signals from the Adjalley et al 2016 data (originally cited reference 16) and data from 5′ cap binding protein-based enrichment (Shaw et al 2021; originally cited reference 17) by concatenating them into one file, i.e., the supporting evidence for 5′ capped ends is the union of the 5′ cap signals from different studies. We revised this section to clarify how the 5′ cap signal information was prepared (line 675–677). Although there is considerable overlap among the 5′ cap signals from these two different data sources, some signals are unique to each study, which could reflect 5′ end sequence biases inherent to each method (Shaw et al 2021; originally cited reference 17). Hence, the 5′ cap signals from the two datasets are complementary and combining them increases the representation of all 5′ capped isoforms expressed during the IDC. 

4. Line 207-208: "higher percent GC (Fig 5A) and shorter length (Fig 5B) compared with spliced introns; however, it

should be noted that the mean differences are small." Please add numbers and appropriate tests to support these statements.

Response: the details of statistical tests performed and P-values were included in the Fig 5 legend. For clarity, we added that Kolmogorov-Smirnov tests were performed to the body of the text (line 209). 

5. Does the analysis of ATSS and ATTS clusters change if only the consensus isoforms (isoforms with support from other studies) are used

Response: We understand the reviewer’s concern that the overall patterns of ATSS and ATTS could be different if only the subset of isoforms with support from all independent studies was used, as the full isoform catalog is biased by the much larger body of data from our study compared with others. To address this concern, we defined consensus isoforms as the subset of isoforms with support from all four sources of long-read RNA-seq data (Yang et al PacBio data, PacBio and Nanopore cDNA data from our study, and the Lee et al direct RNA data). Data support for each isoform was determined using the TAMA_merge_trans.txt output (blue headed columns) in S2B Table to identify the consensus isoforms. We then repeated the analysis of ATSS and ATTS using the 779 consensus isoforms (see figure below). The consensus isoforms encompass 645 different ATSS positions with associated H2A.Z occupancy data. The distribution of these epigenetic data shows a similar bimodal pattern as the full ATSS dataset. We calculated the first two principal component (PC) scores from the H2A.Z occupancy data. We then performed unsupervised clustering of these positions using the PC scores with the CEC R package under the same settings as used for all ATSS, which resolved two clusters. The patterns of base composition of flanking genomic sequence are similar to the C1 and C2 clusters in the full isoform dataset, in which the largest cluster shows elevated frequency of T upstream of the ATSS position, whereas the smaller cluster shows elevated frequency of A and G downstream of ATSS. The majority of positions in the largest cluster are also located outside of annotated protein coding regions. 

With regard to ATTS, the consensus isoforms encompass 546 different ATTS positions. The pattern of base composition of ATTS flanking genomic sequence is similar to the full dataset. We conclude that the patterns of ATSS and ATTS are essentially unchanged when using the consensus isoforms.

Figure response to comment 5 from reviewer #3.

779 consensus isoforms were identified with long-read RNA-Seq data support from four sources (Yang et al 2021 PacBio (at least one of two sample replicates), Lee et al 2021 Nanopore direct RNA, PacBio data from this study (at least one sample replicate), and Nanopore cDNA from this study). Part (A) shows the distribution of natural log normalized H2A.Z histone occupancy (data from Bartfai et al 2010; data were obtained from 10, 20, 30 and 40 hours post-invasion [hpi]) of 640 genomic positions corresponding to isoform 5′ ends as shown by kernel density plot, in which the vertical axis is the probability density function of the variable (natural log normalized H2A.Z histone occupancy). (B) Average base composition of genome sequence flanking 100 bp on either side of consensus isoform 5′ ends for the cluster C1 (486 positions) and C2 (154 positions) assigned from H2A.Z histone occupancy data. (C) Location of C1 and C2 consensus isoform 5′ ends with respect to annotated protein-coding sequence (CDS) regions. (D) Average base composition of genome sequence flanking 100 bp on either side of alternative transcript termination sites (ATTS) for consensus isoforms (546 positions). 

6. An isoform specific qPCR is a must to validate the results which could be done on retained introns isoforms or stage specific isooforms.

Response: We appreciate the reviewer’s concern for validation of isoforms with retained introns. However, we already validated the presence of retained intron events using short-read RNA seq data from the Chappell et al 2020 study (cited reference 15) as shown in Fig. 7A. These data were not used to construct the isoform catalog and thus are qualified as independent data for validation. Short-read RNA-Seq data are suitable for detecting retained intron events (reviewed by Broseus and Ritchie 2020; DOI: 10.1016/j.csbj.2020.02.010). We did not attempt to quantify differences in the abundance of isoforms with retained intron across the IDC. Instead, we clustered retained intron events based on their expression pattern throughout the IDC. We agree with the reviewer that reverse-transcription quantitative PCR (RT-qPCR) with isoform-specific primers can quantify relative isoform abundance including that of retained intron isoforms, albeit with the limitation to two mutually exclusive isoforms differing by a single alternative splicing event (Camacho Londoño and Philipp 2016; DOI: 10.1186/s12867-016-0060-1). This condition does not apply to many genes in which we identified multiple isoforms carrying the same retained intron event and multiple isoforms carrying the intron-spliced event. As an example below, we show isoforms assigned to the FPPS/GGPPS gene (PF3D7_1128400) with 11 annotated introns.

Figure response to comment 6 from reviewer #3.

Screenshot from IGV software showing PlasmoDB annotated genes (track labeled PlasmoDB-57_Pfalciparum3D7.gff) and catalog annotated isoforms in the track labeled S1_Dataset.bed. Reference transcripts are in blue and novel isoforms are in red. The color hue reflects the level of support from long-read RNA-Seq data (darker, more support). 

As shown in the figure above, there are three isoforms with retention of intron 2 (G3932.10, G3932.14 and G3932.16). Isoform G3932.16 also has retention of intron 3. Isoforms G3932.11 and G3932.12 (reference transcript in blue) show splicing of intron 2. Intron 3 is spliced in isoforms G3932.10, G3932.11, G3932.12, and G3932.14. The retained intron events in these isoforms were reported in the study by Gabriel et al 2015 (DOI: 10.1038/srep18429), which employed pyrosequencing of gene-targeted RT-PCR products. In the revised manuscript, we reworded the results and discussion to avoid claims of quantitative differences in expression or abundance at different stages of development (line 232−233; 380−382). We highlighted the lack of quantitative data for isoform abundance in the “limitations of the study” sub-section of the revised Discussion (line 445–450). 

7. The authors state that both PacBio and Nanopore sequencing was used, but a comparison between of data generated using the two techniques is currently missing. Given that the data is not always concordant such an analysis is paramount to generate a viable annotation.

Response: The reviewer has raised an important issue, namely possible discordancy of candidate isoforms generated from PacBio data compared with those from Nanopore data owing to the differences in base-calling and alignment accuracy among the two sequencing platforms. To obtain the most accurate isoform annotation with respect to exon boundaries, PacBio data should be given priority over the less accurate Nanopore data. The isoform catalog was constructed by merging candidate isoforms assigned separately from PacBio and Nanopore data in which PacBio data were given priority over Nanopore data for assigning the merged isoform 5′ terminus and internal splicing junctions (original manuscript line 653−657). Direct RNA Nanopore data were given equal priority with PacBio data for assigning the 3′ terminus as direct RNA is not prone to internal priming artifacts (Balázs et al 2019; cited reference 33). As shown in S2B Table, the vast majority of novel isoforms (99.6%) have support from PacBio data. Only 55 novel isoforms have support from Nanopore data (direct RNA and cDNA data), but with no PacBio data support. Hence, the structures of most isoforms in the catalog were inferred with support from the more accurate PacBio data. In the revised manuscript, we added to the results section with reference to S2B Table that exon junctions for most isoforms are defined using data from the more accurate long-read RNA-Seq platform (line 139−141).

Minor points

1. In multiple places the statement "we utilized long-read platforms to sequence cDNA of P. falciparum IDC stages" is used. This should mention what platforms and which stages clearly. Also, the number of samples (replicates) read depths, coverage and other sequencing QC information should be clearly mentioned in the text.

Response: We wish to clarify that the statement, “we utilized long-read platforms to sequence cDNA of P. falciparum IDC stages” was used once at the end of the Introduction section (line 87). We prefer not to distract the reader with experimental details in the Introduction. The summary statistics for all sequenced samples are provided in S1 Table, which was cited in the body of the text in the Results section (line 132–134). As suggested by the reviewer we added details of sequencing platform, sampled stage of development and RNA fraction to the S1 Table and mentioned in the text what details are included in S1 Table (revised manuscript line 130– 133). 

2. Line 141: "9,004 of the 12,495 novel isoforms were supported by data from other studies in combination with data from this study", supported by atleast one oher study or all the studies

Response: changed to “at least one other study”

3. Line 155-158: Please indicate the percentages of different events in the text as well.

Response: we added the percent of total isoform events for the major categories (ATSS, ATTS and RI) to the text for clarity as suggested by the reviewer (revised manuscript line 155–159). 

4. Line 166: Please give reference of the H2A.Z dataset

Response: The H2A.Z dataset was cited in the Fig. 3 legend (Bártfai et al 2010; originally cited reference 32) and in the Materials and methods section (line 711–712). We added the citation of this reference to the results section as well for clarity (revised manuscript line 165). 

5. This study will be extremely useful to the community if an updated gtf file is provided with new isoforms along with the level of support for it ( ex. supported by 2 studies, 3 studies etc)

Response: we agree that it would be useful for the isoform annotation file to show the level of support so that the reliability/reproducibility of different isoforms for each gene can be easily visualized and compared. In the revised manuscript, we replaced the S1_Dataset.gtf with a new isoform annotation data file in .bed12 format (S1_Dataset.bed). The new file contains an arbitrary score for each isoform which reflects the level of support from different datasets shown in the S2 Table. The scores can be displayed as different color hues in genome browser software such as IGV, i.e., darker hues indicate greater level of support. Moreover, we colored reference transcripts in blue and novel isoforms in red so that reference transcripts can be distinguished from novel isoforms (related to comment 2 by reviewer #1). 

6. Manuscript should have a data availability statement.

Response: all raw sequence data are already deposited in the NCBI GEO under accession number GSE186109, as indicated in the “Data Availability statement” in page 5 of the original manuscript file (appears above the Cover letter). The raw sequencing data in the accession (SRA files) will be made publicly available upon acceptance of the manuscript.

Reviewer #4: Shaw et al., in the current manuscript have catalogued additional alternative splicing (AS) events in P. falciparum IDC stages using the long read next generation sequencing technologies like PacBio and Nanopore platforms. Of the 12,495 novel mRNA isoforms detected by them, majority were either alternative 5' or 3' events. They have also identified a good number of intron retention events. This study further provides information about the in-silico identification of additional proteins that adds to the current proteome repertoire. Long read sequencing technologies identify AS events more accurately and hence this study provides important information related to unidentified AS events that can assist in understanding gene and protein regulation in the deadly malaria parasite P. falciparum.

Minor comments:

1. In case of intron retention events that were reported, it would have been great to validate a couple of intron retention events using reverse transcriptase quantitative PCR by targeting primers to the retained intron region.

Response: We appreciate the reviewer’s concern for validation of retained intron events. However, we already validated the presence of these events using independent short-read RNA seq data from the Chappell et al 2020 study (cited reference 15) as shown in Fig. 7A (see response to similar comment 6 by reviewer #3).

2. I understand that alternative ORFs can code for variant proteins that have not been annotated yet. I will be happy to see if the authors can validate a few intergenic isoforms for their ability to code proteins. This may also include a couple of isoforms that span two or more genes.

We thank the reviewer for this suggestion and agree that additional data on the ability of alternative ORFs (altORFs) to code proteins would be useful. Ribosome profiling (ribo-seq) is an established method for assessing ribosome engagement and is a proxy for protein synthesis (Ingolia et al. 2009; DOI: 10.1126/science.1168978). We re-analyzed the P. falciparum ribo-seq data published in Caro et al 2014 (cited reference 10) to assess the protein-coding potential of novel isoforms. We determined the translational efficiency (TE) and ribosome release scores (RRS) (Gutmann et al 2013; DOI: 10.1016/j.cell.2013.06.009) for altORFs among intergenic and antisense isoforms. We did not consider altORFs in polycistronic or other isoforms spanning annotated genes as the ribo-seq signals could not be distinguished from ORFs in overlapping reference transcripts. The TE and RRS scores are shown in the revised Fig. 8B and statistical tests of distributions are shown in a new S6 Table. In agreement with our original conclusion from protein sequence analysis, most altORFs in intergenic and antisense isoforms do not appear to be protein coding as their TE and RRS distributions are significantly different from annotated protein ORFs in reference transcripts. Details of the ribo-seq data analysis are provided in a new Materials and methods sub-section (line 790–832), and text was added to revised Results (line 266–284) and Discussion (line 399–401) sections. 

6. PLOS authors have the option to publish the peer review history of their article (what does this mean?). If published, this will include your full peer review and any attached files.

Do you want your identity to be public for this peer review? For information about this choice, including consent withdrawal, please see our Privacy Policy.

Reviewer #1: No

Reviewer #2: No

Reviewer #3: No

Reviewer #4: Yes: AMIT KUMAR SUBUDHI

---

## [Decision Letter · Decision Letter 1]

18 Oct 2022

Transcriptomic complexity of the human malaria parasite Plasmodium falciparum revealed by long-read sequencing

PONE-D-22-13960R1

Dear Dr. %Shaw%,

We’re pleased to inform you that your manuscript has been judged scientifically suitable for publication and will be formally accepted for publication once it meets all outstanding technical requirements.

Kind regards,

Arnab Pain, Ph.D.

Academic Editor

PLOS ONE

Additional Editor Comments (optional):

Reviewers' comments:

Reviewer's Responses to Questions

**Comments to the Author**

1. If the authors have adequately addressed your comments raised in a previous round of review and you feel that this manuscript is now acceptable for publication, you may indicate that here to bypass the “Comments to the Author” section, enter your conflict of interest statement in the “Confidential to Editor” section, and submit your "Accept" recommendation.

Reviewer #3: All comments have been addressed

2. Is the manuscript technically sound, and do the data support the conclusions?

Reviewer #3: Yes

3. Has the statistical analysis been performed appropriately and rigorously? 

Reviewer #3: Yes

4. Have the authors made all data underlying the findings in their manuscript fully available?

Reviewer #3: Yes

5. Is the manuscript presented in an intelligible fashion and written in standard English?

Reviewer #3: Yes

6. Review Comments to the Author

Reviewer #3: (No Response)

7. PLOS authors have the option to publish the peer review history of their article (what does this mean?). If published, this will include your full peer review and any attached files.

Reviewer #3: No
